# Seizure protein 6 controls glycosylation and trafficking of kainate receptor subunits GluK2 and GluK3

Martina Pigoni[1,2], Hung-En Hsia[1,2] (iD), Jana Hartmann[3], Jasenka Rudan Njavro[1,2], Merav D Shmueli[1,2,4] (iD), Stephan A Müller[1,2], Gökhan Güner[1,2], Johanna Tüshaus[1,2], Peer-Hendrik Kuhn[1,2,5], Rohit Kumar[1,2,6], Pan Gao[6,7], Mai Ly Tran[8,9], Bulat Ramazanov[9], Birgit Blank[9], Agnes L Hipgrave Ederveen[10], Julia Von Blume[8,9], Christophe Mulle[11], Jenny M Gunnersen[12,13], Manfred Wuhrer[10] (iD), Gerhard Rammes[14], Marc Aurel Busche[3], Thomas Koeglsperger[6,7] (iD) & Stefan F Lichtenthaler[1,2,15,16,*] (iD)

## Abstract

Seizure protein 6 (SEZ6) is required for the development and maintenance of the nervous system, is a major substrate of the protease BACE1 and is linked to Alzheimer's disease (AD) and psychiatric disorders, but its molecular functions are not well understood. Here, we demonstrate that SEZ6 controls glycosylation and cell surface localization of kainate receptors composed of GluK2/3 subunits. Loss of SEZ6 reduced surface levels of GluK2/3 in primary neurons and reduced kainate-evoked currents in CA1 pyramidal neurons in acute hippocampal slices. Mechanistically, loss of SEZ6 *in vitro* and *in vivo* prevented modification of GluK2/3 with the human natural killer-1 (HNK-1) glycan, a modulator of GluK2/3 function. SEZ6 interacted with GluK2 through its ectodomain and promoted post-endoplasmic reticulum transport of GluK2 in the secretory pathway in heterologous cells and primary neurons. Taken together, SEZ6 acts as a new trafficking factor for GluK2/3. This novel function may help to better understand the role of SEZ6 in neurologic and psychiatric diseases.

**Keywords** BACE1; GluK2/3; HNK-1; protein trafficking; SEZ6
**Subject Categories** Membrane & Trafficking; Neuroscience
**The EMBO Journal (2020) 39: e103457**

## Introduction

Precise development and maintenance of the nervous system is essential for the function of the brain. Defects in these processes cause various psychiatric and neurological diseases, including neurodevelopmental disorders and Alzheimer's disease (AD), the most common neurodegenerative disorder (Brookmeyer *et al*, 2011; Goodman *et al*, 2017). One protein linked to different brain diseases is seizure protein 6 (SEZ6). SEZ6 is an N-glycosylated type I transmembrane protein with predominant expression in neurons, where it localizes to the somatodendritic compartment, including the cell surface (Herbst & Nicklin, 1997; Gunnersen *et al*, 2007; Pigoni *et al*, 2016). SEZ6-deficient mice revealed basic functions for SEZ6 in

1   German Center for Neurodegenerative Diseases (DZNE), Munich, Germany
2   Neuroproteomics, School of Medicine, Klinikum rechts der Isar, Technical University of Munich, Munich, Germany
3   UK Dementia Research Institute at UCL, University College London, London, UK
4   Department of Immunology, The Weizmann Institute of Science, Rehovot, Israel
5   Institut für Allgemeine Pathologie und Pathologische Anatomie, Technische Universität München, Munich, Germany
6   Department of Neurology, Ludwig Maximilian University, Munich, Germany
7   Department of Translational Brain Research, German Center for Neurodegenerative Diseases (DZNE), Munich, Germany
8   Max Planck Institute of Biochemistry, Martinsried, Germany
9   Department of Cell Biology, Yale University School of Medicine, New Haven, CT, USA
10  Center for Proteomics and Metabolomics, Leiden University Medical Center, Leiden, The Netherlands
11  Interdisciplinary Institute for Neuroscience, University of Bordeaux, CNRS, UMR 5297, Bordeaux, France
12  Department of Anatomy and Neuroscience, University of Melbourne, Melbourne, Vic., Australia
13  The Florey Institute of Neuroscience and Mental Health, University of Melbourne, Melbourne, Vic., Australia
14  Department of Anesthesiology and Intensive Care, Klinikum rechts der Isar, Technische Universität München, Munich, Germany
15  Institute for Advanced Study, Technical University of Munich, Garching, Germany
16  Munich Cluster for Systems Neurology (SyNergy), Munich, Germany
    *Corresponding author. Tel: +49 89 4400 46426; E-mail: stefan.lichtenthaler@dzne.de

nervous system development and maintenance, such as in dendritic branching, dendritic spine dynamics, correct synapse formation and proper synaptic transmission, in particular excitatory postsynaptic responses, as well as long-term potentiation (LTP) (Gunnersen *et al*, 2007; Zhu *et al*, 2018). Several of these functions seem to be mediated by full-length SEZ6, but SEZ6 can also undergo ectodomain shedding (Lichtenthaler *et al*, 2018), similar to the AD-linked amyloid precursor protein (APP). This proteolytic removal of the large extracellular domain of SEZ6 is mediated by the AD-linked protease β-site APP cleaving enzyme 1 (BACE1) and contributes to SEZ6 function in controlling dendritic spine dynamics and LTP in mice (Pigoni *et al*, 2016; Zhu *et al*, 2018). The cleaved, soluble form of SEZ6 (sSEZ6) is released from neurons and is found in CSF, where it is increased in depressed, bipolar, and schizophrenic patients as well as in inflammatory pain and may be a biomarker for monitoring BACE1 activity in clinical trials for AD (Maccarrone *et al*, 2013; Pigoni *et al*, 2016; Roitman *et al*, 2019).

Genetic variants of SEZ6 are linked to psychiatric and neurological disorders. For example, deletions in the SEZ6 ectodomain are found in childhood-onset schizophrenia (Ambalavanan *et al*, 2015). Other genetic variants have been reported in intellectual disability (Gilissen *et al*, 2014), while the rare SEZ6 mutation R615H has been suggested to cause a familial form of AD (Paracchini *et al*, 2018). Consistent with the disease link, SEZ6-deficient mice, which are viable and fertile, display an anxiety- and depression-related behavior (Gunnersen *et al*, 2007). Moreover, they show exploratory, motor, and cognitive deficits, which are more severe in mice lacking not only SEZ6, but also its two homologs SEZ6-like (SEZ6L) and SEZ6L2 (Miyazaki *et al*, 2006; Gunnersen *et al*, 2007; Nash *et al*, 2019).

Despite the fundamental functions of SEZ6 in the brain, the molecular mechanisms through which wild-type and mutated SEZ6 contribute to physiological and pathological processes in the nervous system are not yet well defined. SEZ6 localizes to the somatodendritic surface of neurons. The SEZ6 ectodomain contains three CUB (complement subcomponent C1r, C1s/sea urchin embryonic growth factor Uegf/bone morphogenetic protein 1) and five complement control protein (CCP, also referred to as Sushi or short consensus repeat (SCR)) domains. Both CUB and CCP domains are frequently found in proteins of the complement system (Escudero-Esparza *et al*, 2013; Forneris *et al*, 2016), but also in auxiliary subunits of neurotransmitter receptors, for example LEV-10 for acetylcholine receptors (AChR) and Neto2 for kainate receptors (Gally *et al*, 2004; Zhang *et al*, 2009). The presence of CUB and CCP domains also in SEZ6 suggests a role for SEZ6 in neuronal protein–protein interactions, but potential interaction partners are not yet known.

Here, we tested the possibility that SEZ6 controls levels of proteins or protein complexes at the neuronal surface. Mass spectrometry analysis of the membrane proteome from SEZ6 knock-out (SEZ6KO) neurons demonstrated a selective reduction in kainate receptor (KAR) subunits 2 and 3 (GluK2 and GluK3) at the cell surface, leading to a lowering of kainate currents in hippocampal brain slices. KARs belong to the family of ionotropic glutamate receptors (iGluRs), which also comprise NMDA (*N*-methyl-D-aspartate) and AMPA (α-amino-3-hydroxy-5-methyl-4-isoxazolepropionic acid) receptors. Mechanistically, we report that the extracellular domain of SEZ6 can bind GluK2 and controls the complex N-glycosylation of GluK2 and/or GluK3, which occurs late in the secretory

pathway, as well as the correct transport of GluK2 and/or GluK3 to the plasma membrane. This molecular function of SEZ6 was independent of BACE1 cleavage and, consequently, appears to be mediated predominantly by the full-length, but not the cleaved form of SEZ6. Together, these data reveal an unanticipated function for SEZ6 as a specific regulator of the trafficking of KAR subunits GluK2 and/or GluK3 to the neuronal plasma membrane, thus establishing a new mode of regulation of KAR activity.

## Results

### SEZ6 controls surface levels of GluK2 and GluK3 in neurons

We used an unbiased mass spectrometry approach and investigated whether loss of SEZ6 affects protein levels at the cell surface of primary neurons. To this aim, we used a modified version of the "surface-spanning protein enrichment with click sugars" (SUSPECS) method (Herber *et al*, 2018), which biotinylates complex glycosylated, sialylated proteins residing at the surface and late in the secretory pathway. In this method, neurons are metabolically labeled with a chemically modified, azido group-containing sugar (N-azidomannosamine, ManNAz). The azido group is later used for click chemistry-mediated biotinylation of glycoproteins. After uptake into neurons, ManNAz is converted to azido-sialic acid, which gets incorporated into the glycan chains of newly synthesized cellular proteins, when they traffic through the trans-Golgi network. After 2 days of metabolic labeling, the living neurons were biotinylated with dibenzocyclooctyne (DBCO)-biotin, which covalently reacts with the azido group of the glycoproteins residing at the surface and late in the secretory pathway. Biotinylated proteins were enriched with streptavidin agarose and analyzed by mass spectrometry using label-free protein quantification. Floxed Sez6 cortical neurons were treated at 2 days *in vitro* (DIV2) with lentiviral CRE recombinase or GFP to obtain neurons lacking (SEZ6KO) or maintaining SEZ6 (WT), respectively (workflow in Fig 1A). Metabolic labeling occurred from DIV5 to DIV7. At DIV7, surface proteins were biotinylated, enriched with streptavidin agarose, and analyzed (workflow in Fig 1A). The sample preparation workflow showed generally little variation between samples, as indicated with correlation coefficients of larger than 0.94 between different samples (Appendix Fig S1). Using SUSPECS, SEZ6 was consistently detected on the surface of the WT neurons, and not consistently detected in the SEZ6KO neurons, in line with an efficient Cre-mediated SEZ6KO (Figs 2A and EV1). 3,209 proteins were detected in 3 out of 3 experiments by the SUSPECS analysis, and 571 were glycosylated, according to UniProt annotation (Fig 1B and Dataset EV1). 40% out of all the proteins detected, and 90% of the glycosylated proteins were classified as membrane proteins according to UniProt keywords (Fig 1B), proving that our method efficiently enriched for membrane proteins. Proteins were considered as hits if their protein level in SEZ6KO vs. WT neurons was lower than $\log_2$ ratio(SEZ6KO/WT) = −0.5 (0.7 fold change) or higher than $\log_2$ ratio(SEZ6KO/WT) = 0.5 (1.4 fold change) and if the *P*-value of this change was lower than 0.05. This yielded 23 proteins with significantly changed abundance in the SEZ6KO neurons. All other proteins did not show quantitative changes between SEZ6KO and WT neurons (Fig 1C and Dataset EV1), demonstrating that loss of SEZ6 selectively affected cell

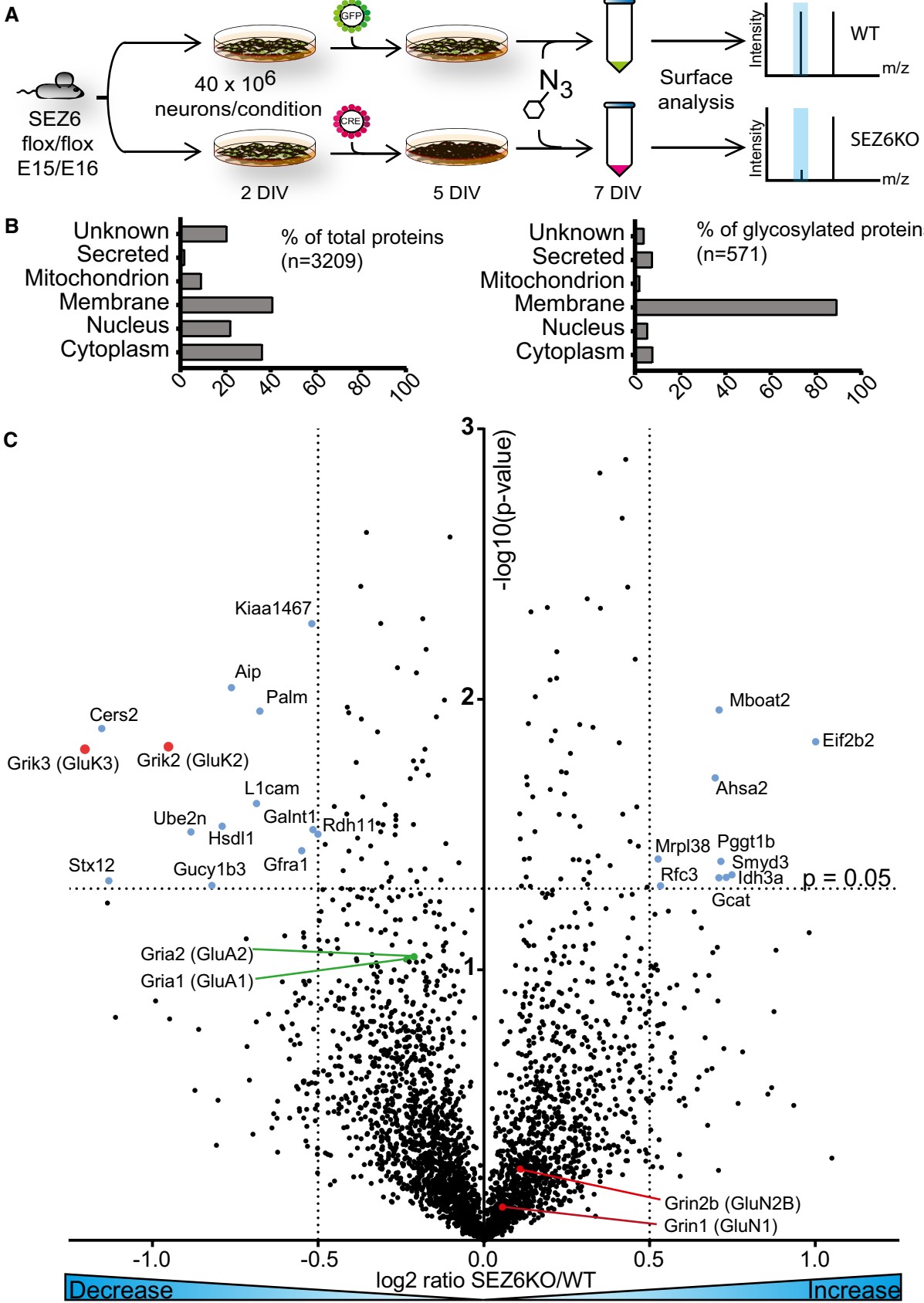

**Figure 1.**

**Figure 1.  Loss of SEZ6 selectively changes the levels of GluK2 and GluK3 at the neuronal surface.**

A  Workflow of the surface analysis performed in SEZ6KO and WT neurons. SEZ6 flox/flox neurons were infected with CRE (SEZ6KO) and GFP (WT) virus at DIV2 and metabolic labeling was started at DIV5. Surface analysis was performed at DIV7.
B  Protein classification of the total proteins (3,209, left panel) and of the glycoproteins (571, right panel) detected in three out of three experiments. More than 90% of the glycoproteins detected are classified as membrane proteins, as expected for the surface labeling.
C  Changes of the 3,209 proteins detected on the cell surface of SEZ6KO neurons compared to WT. The average $\log_2$ fold change of SEZ6KO neurons over WT ($\log_2$ SEZ6KO/WT) is plotted against the negative $\log_{10}$ of each protein's *P*-value according to a one-sample *t*-test. Blue labeled dots are considered as hits (including GluK2 and GluK3) and are proteins with a *P*-value less than 0.05 (neg. $\log_{10}(0.05) = 1.3$; horizontal dotted line) and a fold-change to less than $-0.5$ or more than 0.5 (vertical dotted lines). GluK2 and GluK3 are labeled in pink, GluA1 and GluA2 are indicated with green lines and dots, and GluN1 and GluN2B are marked with red lines and dots.

surface levels of only a few neuronal proteins. Among them were 10 proteins annotated as integral membrane proteins, of which nine had reduced abundance (Grik3, Cers2, Stx12, Grik2, L1cam, Palm, Gfra1, Kiaa1467, and Galnt1) and one increased abundance (Mboat2) in SEZ6 KO neurons. Among the integral membrane proteins with reduced abundance were the two pore-forming subunits of kainate receptors, glutamate receptor kainate 2 (gene *Grik2*, protein GluK2) and 3 (gene *Grik3*, protein GluK3), which showed the strongest reductions at the cell surface of SEZ6KO neurons with a decrease of 50 and 60%, respectively (Fig 1C and Table EV1). Other iGluR subunits were also detected, but not significantly changed (Fig 1C and Dataset EV1). This included two subunits of the NMDA receptor subfamily (GluN1 and GluN2B) and two AMPA receptor subunits (GluA1 and GluA2) (Fig 1C). Thus, among the detected iGluRs, loss of SEZ6 specifically reduced surface levels of GluK2 and GluK3.

Other integral membrane proteins significantly reduced on the cell surface of SEZ6KO neurons were Ceramide synthase 2 ($-50\%$), which is involved in sphingolipid and myelin biosynthesis (Imgrund *et al*, 2009) and Syntaxin-12 (Stx12, also known as Stx13 in humans; $-50\%$), which is an endosomal-trafficking protein contributing to internalization and recycling of various cell surface receptors (Wang & Tang, 2006). Integral membrane proteins with less pronounced surface changes were the neuronal cell adhesion protein L1cam, Paralemmin-1 (Palm), the glial cell line-derived neurotrophic factor (GDNF) family receptor alpha-1 (Gfra1), and the Golgi-localized protein Kiaa1467, which is also known as FAM234B and the glycosylation enzyme N-acetylgalactosaminyltransferase 1 (Galnt1) (Table EV1). The ER-localized lysophospholipid acyltransferase 2 (Mboat2) was the only integral membrane protein with increased abundance in the SEZ6KO. The changed membrane proteins did not have obvious commonalities, and a pathway analysis was not feasible given the low total number of changed proteins. Thus, we focused on the two KAR subunits GluK2 and GluK3, which showed the strongest reduction.

First, we validated the mass spectrometric results using an independent method based on cell surface biotinylation, which selectively labels proteins only at the cell surface, followed by streptavidin pull-down and immunoblotting (Fig EV1A). While surface levels of several transmembrane proteins were not affected in SEZ6KO neurons, such as the SEZ6 homolog SEZ6L2 or members of the glutamate receptor family (GluN2B and GluA2), GluK2 and GluK3 were reduced by 50%, which is consistent with the mass spectrometric analysis (Figs 2A and 1C). Given that the available antibody does not discriminate between GluK2 and GluK3, the protein band is referred to as GluK2/3 throughout the manuscript, in line with previous publications (Zhang *et al*, 2009; Straub *et al*,

2011; Mennesson *et al*, 2019). Interestingly, in addition to the 50% surface reduction, total GluK2/3 protein levels in neuronal SEZ6KO lysates were also reduced, but only by about 15% (see later Fig 4C). Because the surface levels were more strongly reduced than the total levels, the surface/total ratio of GluK2/3 was significantly reduced in the SEZ6KO neurons compared to the WT control (Fig EV1B).

Importantly, no difference in the GluK2 and GluK3 mRNA level was found in SEZ6KO neurons (Fig 2B). Moreover, while mRNA editing at the Q/R site in the channel pore loop of GluK2 is a critical step for controlling cellular protein levels of GluK2 as well as its transport to the cell surface and its degradation (Ball *et al*, 2010; Evans *et al*, 2017), the amount of edited (R) and unedited (Q) mRNA was not altered in the wild-type and SEZ6KO neurons (Fig 2C). Taken together, these experiments suggest that SEZ6 primarily controls cell surface levels of GluK2/3 at the post-translational level, while only moderately affecting total protein GluK2/3 levels.

**SEZ6 controls kainate-evoked KAR currents**

To test whether the reduced surface levels of GluK2 and GluK3 also result in attenuated kainate receptor function, we next recorded kainate-evoked currents in acute hippocampal slices in WT and SEZ6KO mice. Because SEZ6 is not expressed in the CA3 region of the hippocampus (Pigoni *et al*, 2016), we focused instead on the CA1 region, where both SEZ6 and GluK2 are expressed (Bureau *et al*, 1999; Kim *et al*, 2002; Lein *et al*, 2007). CA1 pyramidal neurons were whole-cell patch-clamped, and the membrane current was recorded in control ACSF, in the presence of 10 μM kainate for 5 min and after addition of NBQX, an antagonist of both AMPA and kainate receptors (Fig 3A). Throughout the experiments, AMPA receptors were blocked by GYKI53655 in the ACSF. In addition, the activation of NMDA and GABA$_A$ receptors was prevented by the extracellular presence of APV and bicuculline, respectively. Kainate reliably evoked inward currents in both genotypes. However, in the absence of SEZ6 in the SEZ6KO mice the mean charge carried by the kainate-evoked inward currents (calculated as area under the current curves) was reduced by 34% compared to wild-type mice (Fig 3B), consistent with a reduced membrane abundance of channel subunits GluK2 and/or GluK3.

**GluK2/3 maturation is impaired in SEZ6KO neurons and *in vivo* in mouse brains**

In WT neurons, the GluK2/3 immunoreactivity in Western blots was seen as two closely co-migrating bands, but in SEZ6KO neurons, the upper band appeared reduced and merging with the

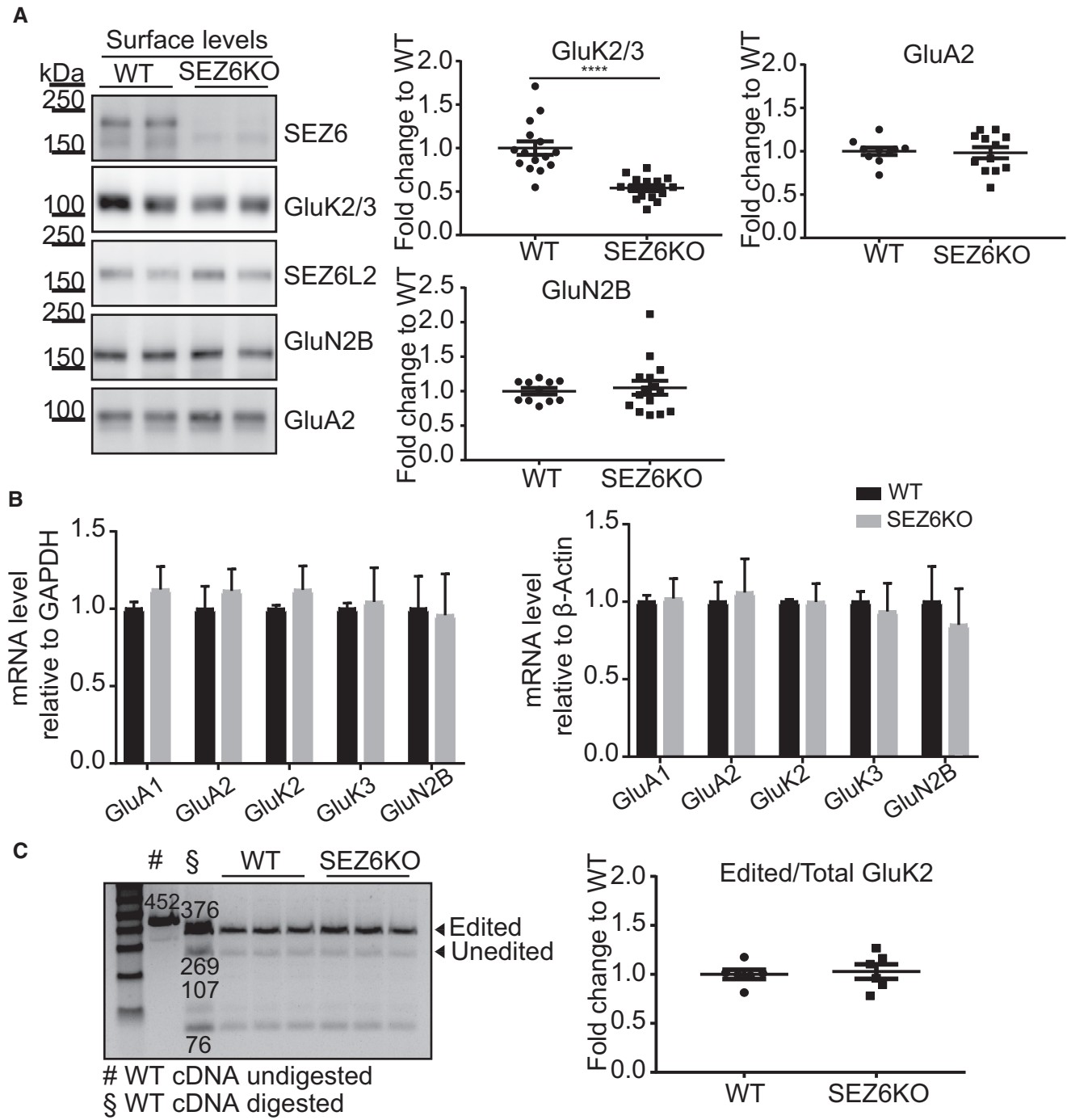

**Figure 2. Validation of GluK2/3 reduction on the surface of SEZ6KO neurons by immunoblot.**

A   SEZ6KO and WT neurons were biotinylated with Sulfo-NHS-Biotin, and surface proteins were enriched by streptavidin bead pull-down. The GluK2/3 antibody cannot discriminate the subunits 2 and 3 (Lerma & Marques, 2013); therefore, the band is commonly indicated with the labeling GluK2/3. GluK2/3, GluA2, and GluN2B surface levels were quantified, normalized to SEZ6L2 surface levels (negative control) in the same sample (GluK2/3/SEZ6L2) and divided by the WT levels, with the ratio for WT being set to 1.0 (plot shows mean ± SEM, at least 10 replicates in 4 independent biological experiments, Mann–Whitney ****$P$-value < 0.005).

B   mRNA levels of GluA1, GluA2, GluK2, GluK3, and GluN2B were quantified in SEZ6KO neurons, normalized to GAPDH and β-actin mRNA in the same sample and compared to the mRNA levels in WT neurons. No difference in GluK2/3 mRNA was detected in SEZ6KO neurons compared to WT (plot shows mean ± SD, 6 replicates in 2 independent biological experiments, multiple $t$-test was used).

C   Editing of GluK2 mRNA was tested in SEZ6KO and WT neurons. The editing value was calculated as (intensity of 376 [edited]/intensity of [376 (edited) + 269 (unedited)]) × 100 and normalized to the band at 76 bp. No difference was detected in SEZ6KO neurons compared to WT (plot shows mean ± SEM, 6 replicates in 2 independent biological experiments, Mann–Whitney test was used).

Source data are available online for this figure.

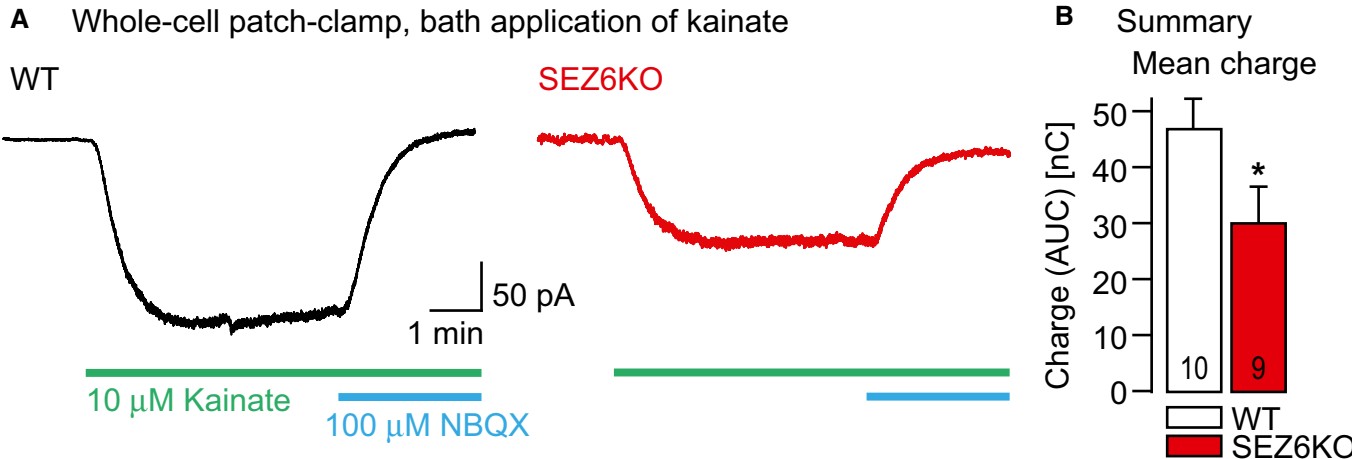

**Figure 3. Reduced kainate-evoked inward currents in the absence of SEZ6.**

A   *Left*: Current trace resulting from whole-cell voltage-clamp recordings at a holding potential of −70 mV in a CA1 pyramidal neuron in a WT mouse hippocampal slice (P15). The current was recorded in the presence of 10 μM bicuculline, 20 μM GYKI53655, and 30 μM APV in the ACSF. Perfusion of the recording chamber with 10 μM kainate and 100 μM 2,3-dioxo-6-nitro-1,2,3,4-tetrahydrobenzo[f]quinoxaline-7-sulfonamide (NBQX) are indicated with green and blue bars, respectively. *Right*: Analogous experiment in a SEZ6KO mouse (P15).

B   Summary of the experiments in A. Bar graph illustrates mean charge with standard error of the mean carried by the inward currents (measured as area under the curve, AUC) for the two genotypes at P15 (WT 46.6 ± 0.5 nC (*n* = 10 cells), SEZ6KO 30.8 ± 0.5 nC (*n* = 9 cells), *P = 0.0375 Mann–Whitney *U*-test).

lower one, suggesting that N-glycosylation of GluK2 and/or GluK3 may be impaired in SEZ6KO neurons (Fig 4A). In fact, GluK2 and GluK3 have multiple N-glycosylation sites (Parker *et al*, 2013; Vernon *et al*, 2017), which are glycosylated to different extents, producing GluK2 and GluK3 forms with more simple, immature glycosylation and with mature glycosylation. GluK2/3 with immature glycans is mostly found within cells, whereas GluK2/3 with mature glycans localizes to a larger extent to the cell surface (Mah *et al*, 2005). Distinction between both sugar modifications is possible with the glycan-removing enzyme endoglycosidase H (EndoH) that cleaves off immature, but not mature N-glycans. To test, whether SEZ6KO affects the glycosylation of GluK2 and/or GluK3, we digested neuronal protein lysates with EndoH and blotted for the GluK2/3 bands (Fig 4A). With EndoH, the two bands of GluK2/3 in

control conditions were converted to three distinct proteins bands with a lower apparent molecular weight. The lowest band is completely deglycosylated, as shown previously by PNGase F treatment (Nasu-Nishimura *et al*, 2010). The upper two bands were not completely deglycosylated, demonstrating that they partially contain mature sugar chains, which are not removable by EndoH. SEZ6KO specifically affected the uppermost, mostly EndoH-resistant protein form. Compared to WT, this band had a lower intensity and was shifted to a slightly lower apparent molecular weight, indicating that in the SEZ6KO, GluK2/3 carries less mature glycans (Fig 4A and model in 4A). To better visualize the differences of the band pattern between WT and SEZ6KO, the intensity distribution of the three bands—as quantified from the immunoblots—was plotted as a histogram (Fig EV2A). The two closely migrating upper bands in

**Figure 4. Loss of SEZ6 impairs glycosylation and maturation of GluK2/3.**

A   In WT neurons, GluK2/3 was seen as two closely comigrating bands, whereas the upper one of lower intensity was missing or running even more closely to the lower band of main intensity in SEZ6KO neurons (schematic representation on the right). When the total lysates of SEZ6KO and WT neurons were digested with endoglycosidase H (EndoH)—which removes immature, but not mature sugars—the two closely comigrating bands were converted to three bands of a lower apparent molecular weight, consistent with full deglycosylation of the lowest band and a partial deglycosylation of the upper two bands (marked in light blue, red, and green in the right panel). In SEZ6KO neurons, the uppermost (light blue) band was missing and a new band of lower apparent molecular weight was seen that was overlapping with the red labeled band and was indicated with the purple asterisk in the SEZ6KO. No difference in the glycosylation of GluA2 was detectable, pointing to a specific effect of SEZ6 on GluK2/3.

B   The running pattern of GluK2/3 in SEZ6KO and WT brains was compared upon EndoH digestion. Similar to primary neurons, GluK2/3 displayed an immature glycosylation pattern in SEZ6KO brains.

C   Total GluK2/3 levels were quantified and showed a significant but moderate reduction in SEZ6KO compared to WT neuronal lysates (plot shows mean ± SEM, 9 replicates in 3 independent biological experiments, Mann–Whitney **P*-value = 0.0012).

D   No significant change in GluK2/3 total levels was detectable in SEZ6KO and WT adult brains (plot shows mean ± SEM, 6 replicates, Mann–Whitney test was used).

E   Synaptosomes from knock-out (KO) brains of single SEZ6 family members (SEZ6, SEZ6L, SEZ6L2), triple knock-out (TKO), and respective controls (WT and TWT—triple WT) were digested with EndoH. GluK2/3 displayed the immature glycosylation in SEZ6KO and TKO brains, but not in SEZ6L and SEZ6L2 brains, demonstrating non-redundant functions for the SEZ6 family members (*n* = 4).

F   Brain homogenates from SEZ6KO and WT mice at different ages were digested with EndoH. GluK2/3 displayed the immature glycosylation at all developmental stages of SEZ6KO mice, suggesting that no compensation effect is occurring with aging.

Source data are available online for this figure.

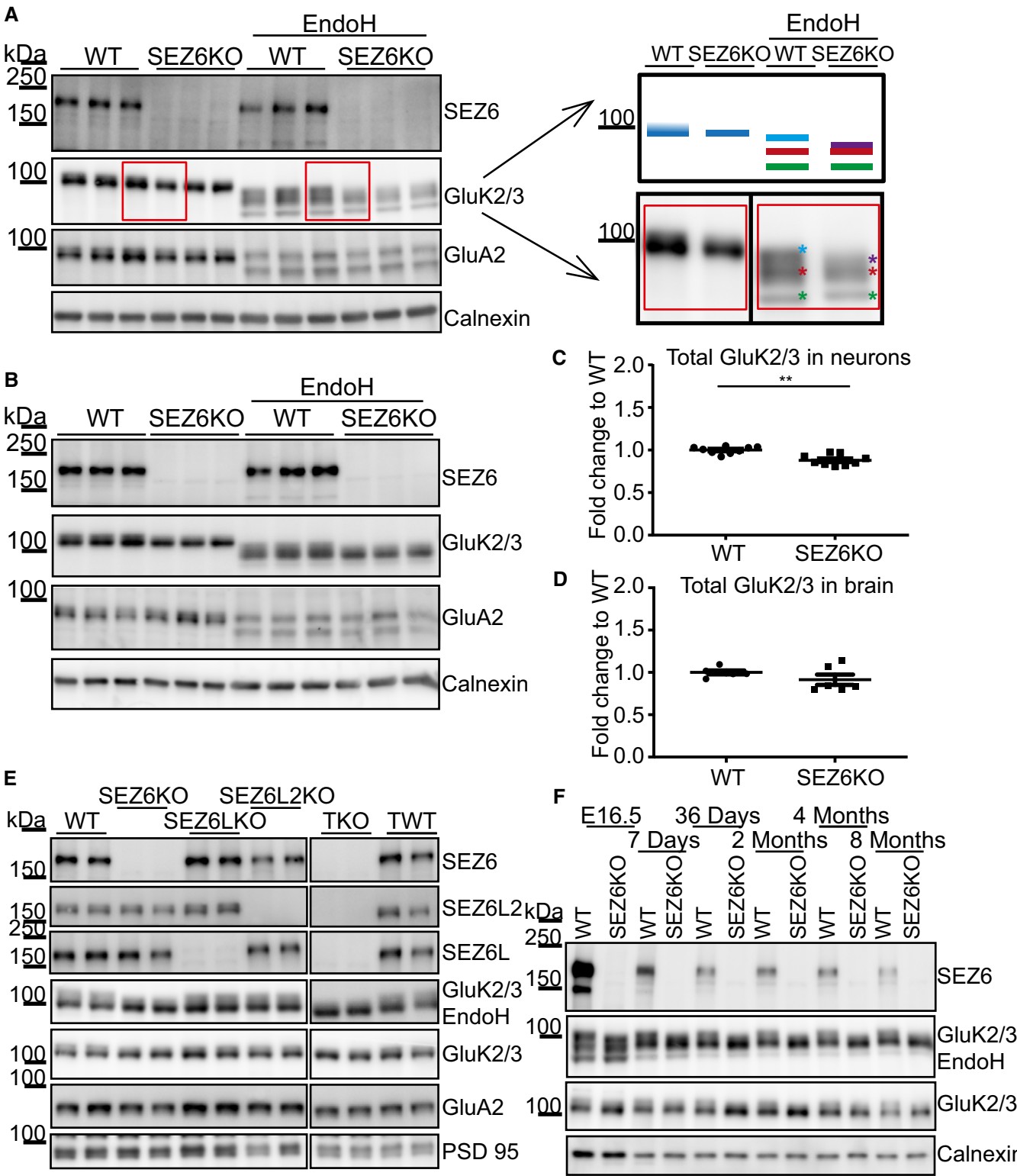

**Figure 4.**

immunoblot of the WT condition are seen as the plateau at the highest intensity in the histogram. In contrast, the disappearance of the highest molecular weight immunoblot band in the SEZ6KO condition resulted in the plateau being converted to a single peak.

Because the upper two bands in the WT are too close to be quantified separately, we quantified the differences between WT and SEZ6KO in another way. We divided the histogram into quartiles and focused on the fourth quartile, corresponding to the highest

molecular weight in the immunoblot, where the differences are seen between WT and SEZ6KO. Mean intensities of the fourth quartile were calculated and were significantly smaller for SEZ6KO compared to WT (Fig EV2A), in agreement with the immunoblot.

So far, our experiments revealed that SEZ6 modulates mature GluK2 and/or GluK3 glycosylation *in vitro* in primary neurons. To test whether maturation of GluK2 and/or GluK3 is also affected *in vivo*, we analyzed their glycosylation pattern in brain homogenates and synaptosomes of wild-type and constitutive SEZ6KO adult mice. Similar to our results *in vitro* (Fig 4A), the upper one of the two GluK2/3 bands under control conditions (no EndoH treatment) was reduced in the SEZ6KO brain and this effect was even more clearly visible after EndoH treatment, where again the uppermost, mature glycoform shifted to a lower apparent molecular weight (Fig 4B for brain homogenates and 4E for synaptosomes and model in Fig EV2B). In contrast to the primary neurons, total levels of the GluK2/3 in the brain samples were not significantly decreased and this was also seen for a control protein, the GluA2 subunit of AMPA receptors (Figs 4B and D, and EV2C).

Although SEZ6 has two homologs, SEZ6L and SEZ6L2, which have a similar domain structure as SEZ6, there was no compensatory change in SEZ6 expression nor an effect on mature glycosylation of the GluK2/3 band in SEZ6L and SEZ6L2 single knock-out brain synaptosomes (Figs 4E and EV2D). Moreover, the reduced maturation of the GluK2/3 band was not further reduced in synaptosomes from triple knock-out mice lacking SEZ6 and both of its homologs (Fig 4E). This demonstrates that specifically SEZ6, but not its homologs, is required for mature glycosylation of GluK2 and/or GluK3.

The relevance of SEZ6 for GluK2/3 maturation was not only seen at very young ages, when SEZ6 expression is high [(Kim *et al*, 2002; Miyazaki *et al*, 2006) and Fig 4F], but also during adulthood (Fig 4F). This demonstrates that SEZ6 has an effect on maturation of GluK2 and/or GluK3 independent of the developmental stage and that no compensation occurs through other proteins during adulthood. Interestingly, the lowest GluK2/3 band was more intense during the embryonic stage, and its intensity decreased sharply after birth (Fig 4F). Considering that the reduction was equally occurring in WT and SEZ6KO mice, we conclude that this effect was dependent on the developmental stage, but independent of SEZ6.

### SEZ6 controls HNK-1 glycan modification of GluK2 and/or 3

To identify the SEZ6-dependent changes in glycosylation of GluK2 and/or GluK3, we used a lectin chip microarray (LecChip) that allows characterization of the glycome fingerprint of proteins (Hu & Wong, 2009). Lysates of WT and SEZ6KO neurons were analyzed by a LecChip containing 45 different lectins, which recognize different sugar structures. This analysis revealed a reduction in glycosylation of GluK2 and/or GluK3 for the lectins PHAE (*Phaseolus vulgaris*-E) and PHAL (*Phaseolus vulgaris*-L) (Fig 5A). These two lectins share the recognition of the oligosaccharide Galβ1-4GlcNAcβ1 as a common motif, a sugar unit commonly involved in complex oligosaccharide formation, including the human natural killer-1 (HNK-1) epitope (Chou *et al*, 1986). This is a known glycan modification of GluK2, which affects GluK2 function (Vernon *et al*, 2017), is found in a few other neuronal proteins, such as GluA2 (Morita *et al*, 2009) and NCAM-1 (Kruse *et al*, 1984) and is generally assumed to control cell adhesion and migration (Morise *et al*, 2017). To determine, whether loss of SEZ6 indeed reduces modification of the GluK2/3 protein band with the HNK-1 glycan, we immunoprecipitated GluK2/GluK3 from mouse brain extracts and blotted with an anti-HNK-1 antibody (Stanic *et al*, 2016 and Fig 5B). In WT brain extracts, the GluK2/3 band contained the HNK-1 modification, in agreement with a previous report (Vernon *et al*, 2017). As a control, treatment of immunoprecipitated GluK2/3 protein with the enzyme peptide N-glycosidase F (PNGase), which removes all N-linked sugars, abolished the HNK-1 signal as expected. Importantly, in SEZ6KO brains, the GluK2/3 protein lacked the HNK-1 modification, while the immunoblot signal for HNK-1 in the total brain extract (without immunoprecipitation of GluK2/3 protein) and for the different isoforms of the HNK-1-modified protein NCAM-1 was unaltered (Fig 5B). Thus, loss of SEZ6 specifically prevents HNK-1 modification of GluK2 and/or GluK3, but not of other proteins in mouse brain. As a further control, global glycome determination of lysates by mass spectrometry did not show significant changes between WT and SEZ6KO neurons (Appendix Fig S2), revealing that loss of SEZ6 does not broadly affect N-glycosylation in neurons.

### SEZ6 facilitates GluK2 transport within the secretory pathway

So far, our experiments revealed that SEZ6 is required for normal cell surface localization and correct glycosylation of GluK2 and/or GluK3 with the HNK-1 epitope, which happens in the trans-cisterna of the Golgi or in the trans-Golgi network (Kizuka & Oka, 2012). These findings suggest that SEZ6 promotes GluK2 and/or 3 protein trafficking through the secretory pathway.

---

**Figure 5. Loss of SEZ6 reduces HNK-1 epitope on GluK2/3.**

A  The glycome fingerprint of GluK2/3 was analyzed by lectin chip microarray (LecChip). Lectin PHAL and PHAE (indicated with asterisks) detected reduced amounts of the oligosaccharide Galβ1-4GlcNAcβ1 on GluK2/3 in SEZ6KO neurons (plot shows mean ± SEM, 3 WT replicates and 4 SEZ6KO replicates were used, discoveries were determined using the FDR method of Benjamini and Hochberg, with Q = 2%).

B  Immunoblot analysis revealed reduction of HNK-1 epitope on GluK2/3 in SEZ6KO brains. *Left panels*: Comparable amounts of GluK2/3 were immunoprecipitated (IP) in WT and SEZ6KO brains using excess of brain homogenates compared to beads. Anti-HNK-1 antibody was used for detection. HNK-1 band on GluK2/3 was detected only in WT brains but not in SEZ6KO brains (n = 6). As a control, HNK-1 modification of NCAM-1 was unaltered, as revealed after IP of NCAM-1 from WT and SEZ6KO brains (n = 3), followed by detection with anti-HNK-1 antibody. Different isoforms of NCAM-1 are annotated with arrowheads (NCAM 180, NCAM 140, and NCAM 120). *Mid panel*: To prove the specificity of the HNK-1 antibody, GluK2/3 was immunoprecipitated and digested with Peptide-N-Glycosidase F (PNGase F), which removes N-linked oligosaccharides. Upon PNGase F digestion, HNK-1 was not detectable in WT brains and the molecular weight of GluK2/3 was reduced (n = 2). *Right panels*: In total brain homogenates (input), no difference of general HNK-1 epitope or NCAM-1 levels was detected.

Source data are available online for this figure.

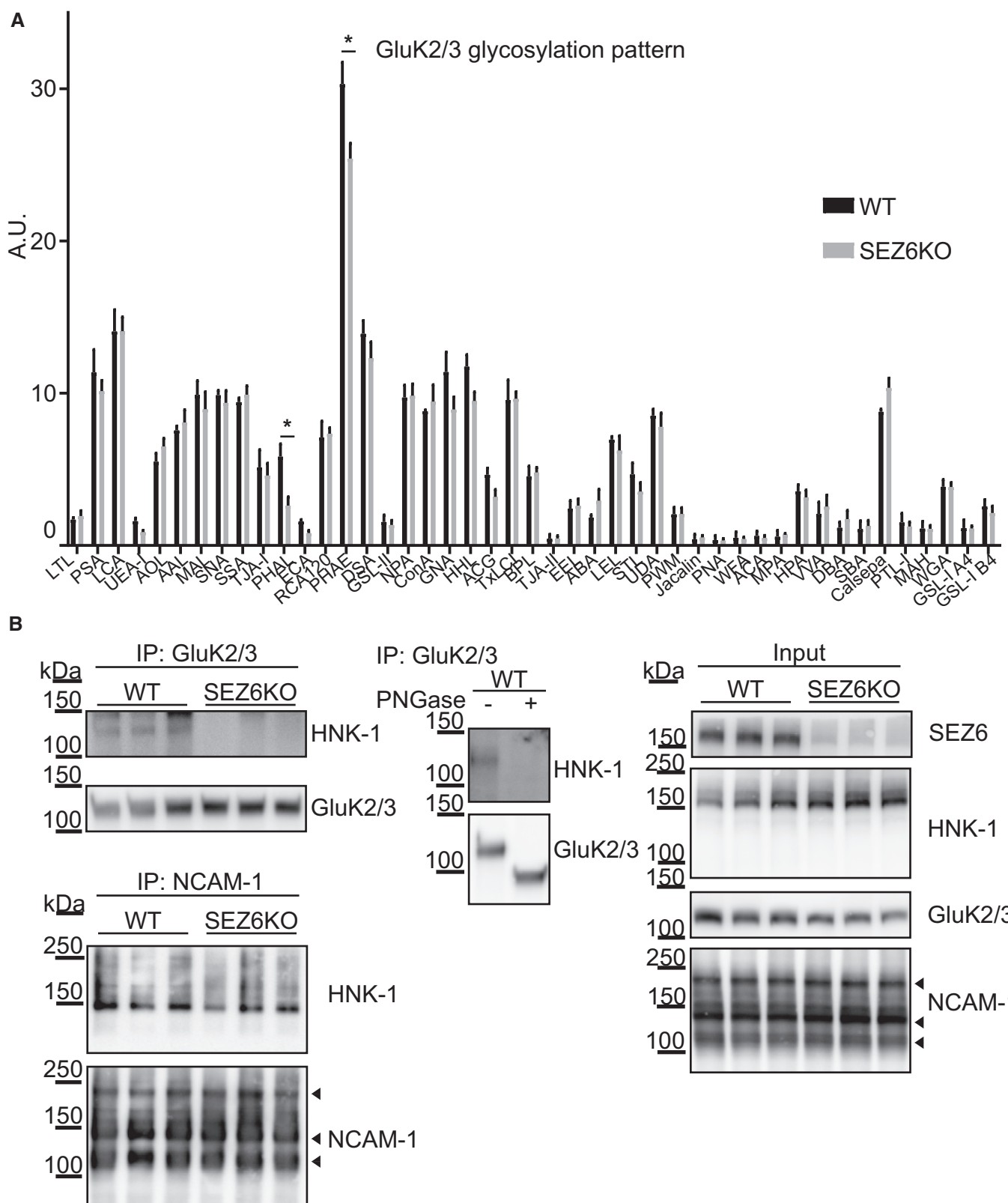

**Figure 5.**

To investigate whether SEZ6 directly affects GluK2 trafficking through the secretory pathway towards the plasma membrane, we used the "retention using selective hooks" (RUSH) system (Fig 6A). Human embryonic kidney 293 (HEK293) cells were chosen as a heterologous system, which do not express endogenous SEZ6 or GluK2 or GluK3, and therefore, no competition with endogenous proteins is expected. HEK293 cells were transfected with a GluK2 construct (SBP-GFP-GluK2) that is retained in the endoplasmic reticulum (ER) unless biotin is added to the culture medium, which binds to a streptavidin-KDEL ER anchor and suppresses the interaction with a streptavidin-binding peptide (SBP) at the N-terminus of GluK2 (Fig 6A). Consequently, GluK2 is released from the artificial ER anchor. Thus, the release of SBP-GFP-GluK2 from the ER can be induced in a synchronous manner (Boncompain *et al*, 2012; Evans *et al*, 2017). In the absence of added biotin (0 min), SBP-GFP-GluK2 was retained in the ER as shown by colocalization with the ER marker Calnexin (Fig EV3A top panels). 20 min after biotin addition, SBP-GFP-GluK2 largely localized to the Golgi apparatus as shown by colocalization with the Golgi markers GM130 (second-panel row) and TGN46 (third-panel row), respectively. Localization in vesicles was also observed at this time point. After 40 min of biotin addition, vesicles containing SBP-GFP-GluK2 were observed in the cytoplasm of the cells in addition to a TGN colocalization. To determine the identity of these vesicles, SBP-GFP-GluK2 containing cells were co-transfected with SBP-LysozymeC-RFP (SBP-LysC-RFP), a marker for sphingolipid-rich TGN-derived vesicles (Deng *et al*, 2018). As shown in Fig EV3A (lowest panel row), LyzC vesicles colocalized with SBP-GFP-GluK2 vesicles after 40 min of biotin addition. These data suggest that both proteins are transported in the same vesicles to the cell surface. Together, these data demonstrate that the SBP-GFP-GluK2 system represents a robust assay to monitor the trafficking of GluK2 in living cells.

Next, we investigated the impact of transfected SEZ6 on SBP-GFP-GluK2 trafficking. As a control, we used a mutant form of SEZ6 where the cytoplasmic tail was replaced by an ER retention motif, which prevented the mature glycosylation of SEZ6 (see Fig 8A and D for schematic overview and immunoblot), consistent with its ER retention and its reticular staining observed by immunofluorescence analysis (Fig 6B). Importantly, compared to the ER-retained SEZ6ΔcytoER mutant, full-length wild-type SEZ6 (SEZ6FL) resulted in a significant increase in TGN-derived vesicles both after 20 (mean number of vesicles/cell: $25 \pm 8$) and 40 min of biotin addition (Fig 6C and D, mean number of vesicles/cell: $36 \pm 11$). Similar results were obtained for SEZ6FL when another protein (inactive Cre) different from SEZ6ΔcytoER was used as negative control (Fig EV3B). These experiments reveal that SEZ6 promotes SBP-GFP-GluK2 trafficking.

The RUSH data in Fig 6C showed a partial overlap of the SEZ6 and SBP-GFP-GluK2 staining, suggesting that SBP-GFP-GluK2 and SEZ6 colocalize in vesicles. To determine to which extent these vesicles overlap, we analyzed the percentage of colocalizing TGN-derived vesicles in cells expressing SBP-GFP-GluK2 and SEZ6ΔcytoER (negative control) or SEZ6FL or SBP-LyzC-RFP (positive control), respectively, after 40 min after biotin addition. No significant colocalization was observed for SBP-GFP-GluK2 and the negative control SEZ6ΔcytoER, whereas a significant increase to 20 and a 50% colocalization was observed for SBP-GFP-GluK2 with SEZ6FL and SBP-LyzC vesicles, respectively (Figs EV3C and 5D). From these

data, we conclude that SBP-GFP-GluK2 and SEZ6 are partially cotransported from the TGN to the cell surface, further supporting the function of SEZ6 in promoting trafficking of GluK2 to the cell surface.

Next, we used the RUSH system to confirm the role of SEZ6 for GluK2 trafficking under a more physiological condition in primary hippocampal neurons, where SEZ6 is endogenously expressed (Fig 7). We used a similar RUSH construct for GluK2 as in the HEK cells, but GFP was replaced by mCherry (SBP-mCherry-GluK2). We validated that after incubation with biotin, SBP-mCherry-GluK2 started to be transported to the neuronal cell surface, as evidenced by surface labeling of SBP-mCherry-GluK2 with the SBP antibody, which was added to the living cells together with biotin (Fig 7A and B). The surface/total ratio of SBP-mCherry-GluK2 constructs, estimated by measuring the ratio of SBP/mCherry fluorescence signals, significantly increased after 20 min of incubation with biotin ($0.49 \pm 0.08$) as compared to the no biotin control ($0.15 \pm 0.04$). After 40 min of incubation with biotin, the ratio further increased to close to one ($0.90 \pm 0.08$), indicating that in primary neurons, most of the SBP-mCherry-GluK2 constructs have been transported to the cell surface after 40 min with biotin. Therefore, we compared the SBP-mCherry-GluK2 surface/total ratio in control and SEZ6 KO primary neurons after 20 min of incubation with biotin, when many of the constructs have not yet completed their transported to the surface. In SEZ6 KO primary neurons, the SBP-mCherry-GluK2 surface/total ratio was significantly reduced (Fig 7C and D), which is in line with the results obtained in HEK293 cells. Taken together, we conclude that SEZ6 influences GluK2 maturation by promoting its trafficking through the secretory pathway.

## The extracellular domain of SEZ6 binds GluK2/3 and is required for mature glycosylation of GluK2/3

To further address the mechanism through which SEZ6 promotes trafficking, glycosylation, surface localization, and activity of GluK2 and/or GluK3, we tested which of the several SEZ6 domains is required and whether SEZ6 interacts with GluK2 and/or GluK3.

The SEZ6 ectodomain contains CUB and CCP domains, which may mediate protein–protein interactions, whereas its cytoplasmic domain includes an NPxY motif, which is known to be important for endocytosis of proteins from the cell surface (Bonifacino & Traub, 2003). To investigate which of the protein domains is required for full maturation of GluK2 and/or GluK3, we carried out a domain-deletion analysis of SEZ6. SEZ6KO cortical neurons were infected the same day they were plated (DIV0) with the lentiviral constructs represented in Fig 8A, and lysates were collected at DIV7 for glycosylation analysis. As a control, we used full-length wild-type SEZ6 (SEZ6FL) (Fig 7A). First, one mutant lacked the C-terminal 39 amino acids (SEZ6ΔCyto, amino acids 20–952; amino acids 1–19 are the signal peptide) including the NPxY motif. Second, another mutant was the SEZ6 ectodomain (SEZ6ecto, amino acids 20–909), which has the same amino acid sequence as the soluble SEZ6 (sSEZ6) generated through BACE1 cleavage between leucine906 and aspartate907 of SEZ6 (Pigoni *et al*, 2016), but contains additional HA and Flag tags. The third mutant was the counterpart to SEZ6ecto, namely the C-terminal fragment (SEZ6CTF, amino acids 907–991) generated by BACE1 cleavage. First, we verified that the constructs were correctly expressed in the SEZ6KO

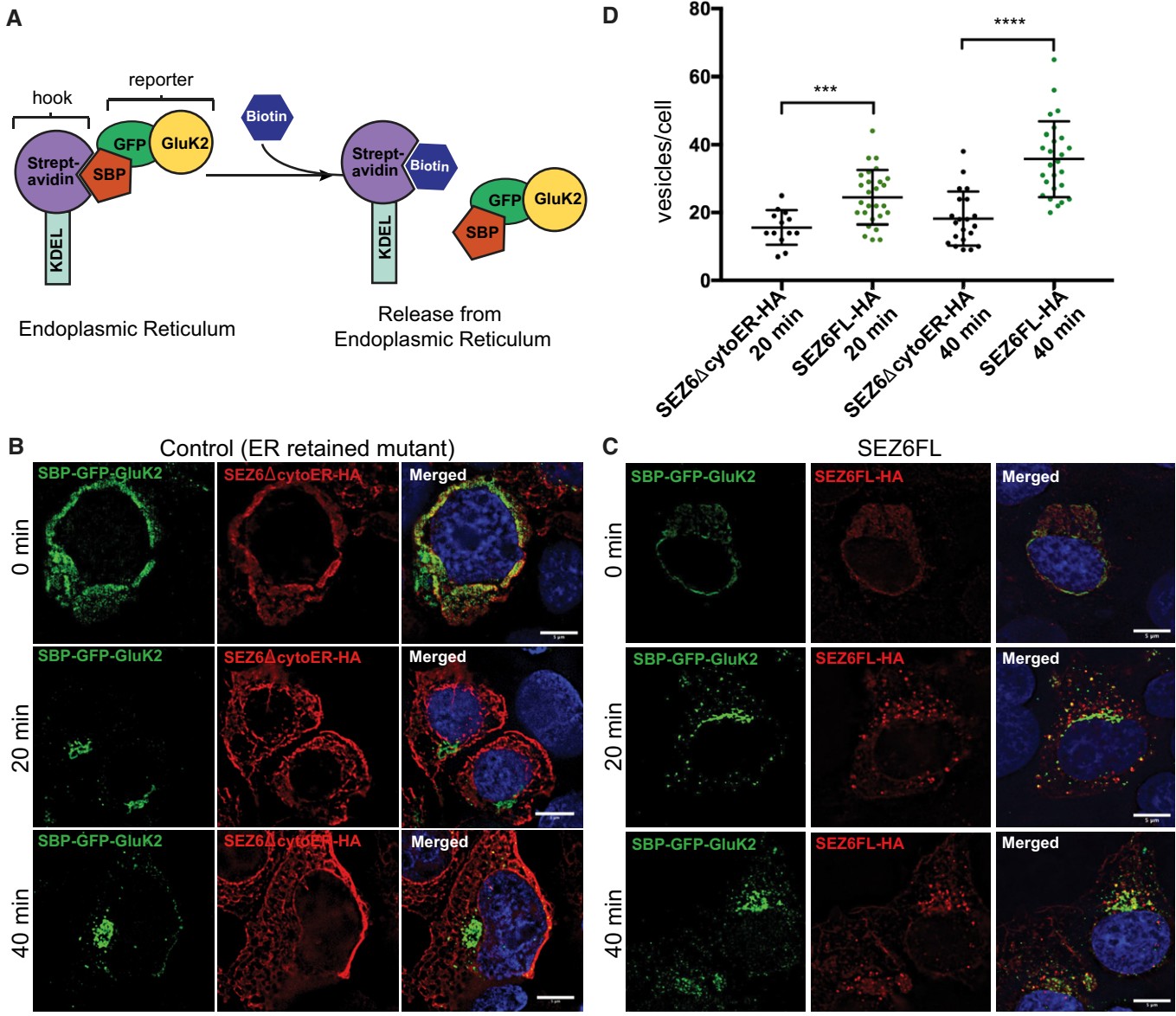

**Figure 6. RUSH assay to analyze trafficking of SBP-GFP-GluK2 in HEK293T cells.**

A    Schematic representation of the RUSH Cargo Sorting Assay using confocal microscopy. HEK293T cells were transfected with the SBP-GFP-GluK2 plasmid that expresses a fusion protein consisting of streptavidin binding peptide (SBP) followed by GFP and GluK2, and a streptavidin-KDEL "anchor." The ER retention signal KDEL retains SBP-containing proteins in the ER. Upon addition of biotin to the cell culture medium, SBP-GFP-Gluk2 is released allowing its trafficking through the secretory pathway.

B, C    The constitutively ER-retained SEZ6 mutant SEZ6ΔcytoER-HA (B) and full-length wild-type SEZ6FL-HA (C) were co-transfected with SBP-GFP-GluK2. Biotin was added to elicit release of SBP-GFP-GluK2 from the ER (0 min), and cells were fixed at different time points (0, 20, and 40 min). HA-tagged SEZ6FL and SEZ6ΔcytoER transgenes were labeled with anti-HA (mouse) and anti-mouse-Alexa594 antibodies. Size bars represent 5 μm.

D    Scatter dot plot represents number of vesicles per cell in SBP-GFP-GluK2 RUSH experiment with SEZ6ΔcytoER-HA (control) or SEZ6FL-HA co-transfection. Vesicle counts from 15 to 28 cells per timepoint from 2 independent experiments with mean number of vesicles and error bars (SD) are shown. Mann–Whitney test was used to compare HA-tagged SEZ6FL and SEZ6ΔcytoER at each time point. At 20 min ***$P$-value = 0.0007, at 40 min ****$P$-value < 0.0001.

neurons. SEZ6FL displayed the known double band corresponding to the mature form (upper band) and the immature form of the protein (lower band) (Fig 8B; Pigoni *et al*, 2016). SEZ6ΔCyto showed bands of similar molecular weight, but of lower intensity. SEZ6ecto had the expected lower molecular weight than the mature SEZ6FL and was detected in the lysate as the immature form, whereas the mature glycosylated form was efficiently secreted and

detected in the conditioned medium of the neurons (Fig 8C). SEZ6CTF was well expressed at the expected low molecular weight (24 kDa). SEZ6FL, SEZ6ΔCyto, and SEZ6ecto rescued the glycosylation phenotype, as seen by the reappearance of the uppermost GluK2/3 band after EndoH treatment in the SEZ6KO neurons (Fig 8B), whereas SEZ6CTF did not rescue the maturation of the GluK2/3 protein. We also quantified the re-appearance of the

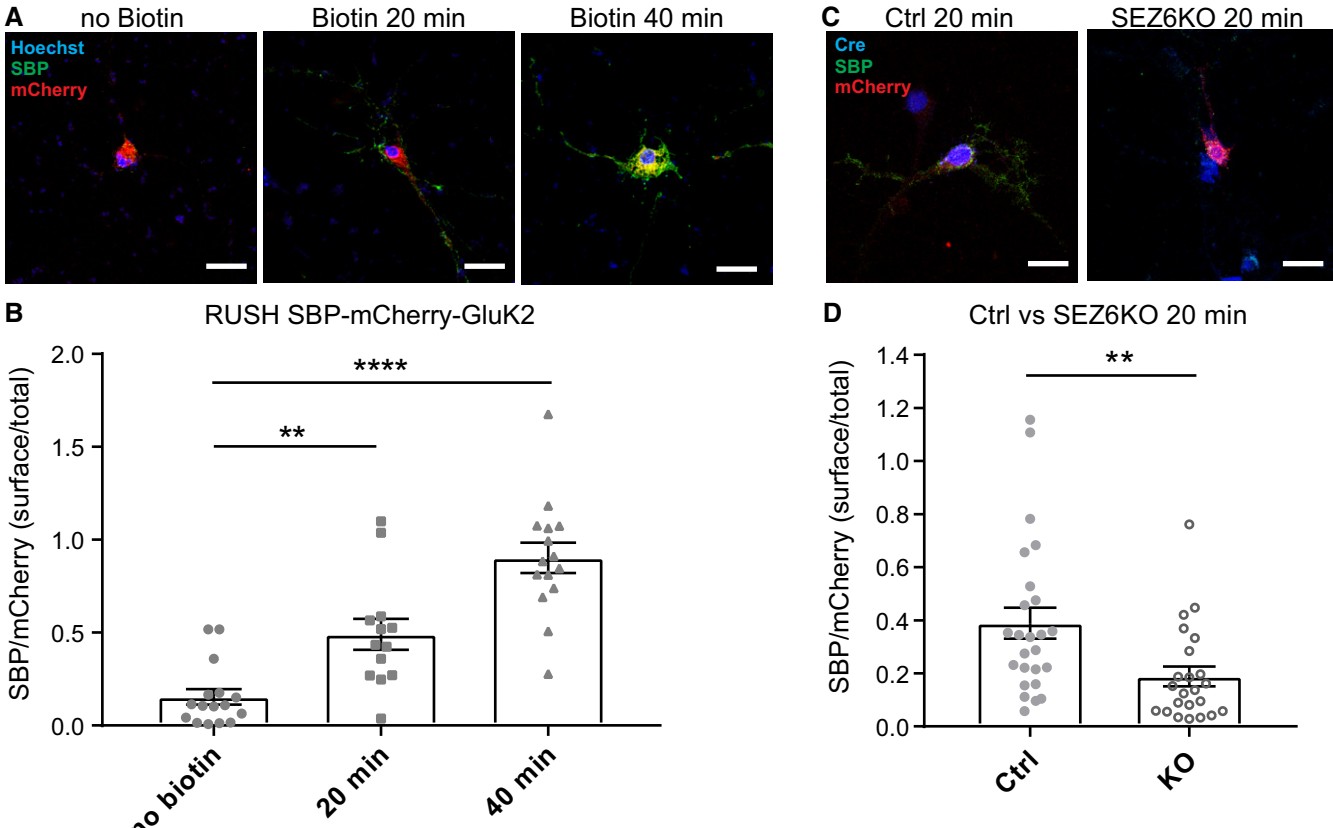

**Figure 7. SEZ6 facilitates GluK2 trafficking in primary neurons.**

A   Representative pictures of WT hippocampal neurons transfected with the SBP-mCherry-GluK2 RUSH construct, followed by treating neurons with biotin and anti-SBP antibodies (mouse, green) for 20 min or 40 min to live-label the GluK2 transported to the neuronal cell surface. As a control (no biotin), neurons were incubated only with SBP antibody for 40 min. The surface/total ratio of SBP-mCherry-GluK2 constructs was estimated by calculating the ratio of SBP (green)/mCherry (red) fluorescence signals in the same area chosen for quantification.

B   Quantification results for (A), indicating that incubation with biotin increases the surface/total ratio of SBP-mCherry-GluK2 over time. No Biotin $n = 16$, 20 min $n = 13$, 40 min $n = 15$, all biological replicates. Graph shows mean $\pm$ SEM, 1-way ANOVA $P < 0.0001$ and Dunnett's *post hoc* test, no biotin vs. 20 min **$P = 0.0030$; no biotin vs. 40 min ****$P < 0.0001$.

C, D   SEZ6 flox/flox neurons were lentivirally infected with inactivated Cre (Y324F, Ctrl) or Cre recombinase (SEZ6KO), followed by transfection with the SBP-mCherry-GluK2 construct and treated with biotin and anti-SBP antibodies (mouse, green) for 20 min. The expression of inactivated Cre or Cre recombinase was confirmed by immunostaining with an anti-Cre antibody (rabbit, blue). SEZ6KO neurons showed a significantly reduced surface/total ratio of SBP-mCherry-GluK2 (Ctrl $n = 25$, SEZ6KO $n = 23$ biological replicates). Graph shows mean $\pm$ SEM, unpaired *t*-test, **$P = 0.0068$).

Data information: Scale bars represent 20 µm.

uppermost band as described above (Fig EV2A) by determining the normalized mean intensities in the fourth quartile, which were significantly increased for SEZ6FL ($105.3 \pm 1.4\%$, $P = 0.044$), SEZ6ΔCyto ($105.6 \pm 0.8\%$, $P = 0.030$), and SEZ6ecto ($108.1 \pm 1.0\%$, $P = 0.002$) compared to the GFP (set to $100\%$, $\pm1.8\%$) or SEZ6CTF-transduced ($97.4 \pm 1.7\%$, $P = 0.478$) SEZ6KO neurons. In accordance to this, GluK2 and/or GluK3 on the cell surface were increased about twofold when SEZ6FL was transfected into SEZ6 KO neurons compared to the control transfected with GFP (Appendix Fig S3A and B). Thus, we conclude that the extracellular domain of SEZ6—either in the membrane-bound or soluble form—is required for the function of SEZ6 in controlling GluK2/3 protein glycosylation. Interestingly, similar observations were made for LEV-10 and SOL-1, where not only the full-length, but also the soluble CUB domain-containing soluble ectodomains rescued surface clustering of neurotransmitter receptors in the corresponding knock-out cells (Gally *et al*, 2004; Zheng *et al*, 2006; Gendrel *et al*, 2009; Nakayama & Hama, 2011).

Within the secretory pathway, the ER exit of KARs can be controlled (Jaskolski *et al*, 2004; Contractor *et al*, 2011). Therefore, we used SEZ6ΔcytoER, containing a KKXX ER-retention motif and showed that, similarly to Fig 6 where this construct was not able to increase the trafficking of GluK2, it was also not able to rescue the glycosylation defect of SEZ6KO neurons in contrast to SEZ6Δcyto (Fig 8D). As expected for the ER localization, SEZ6ΔcytoER was only detected as the immature protein band in the immunoblot. This experiment, together with the RUSH in Fig 6, demonstrates that SEZ6 must affect the post-ER trafficking of GluK2.

The finding that expression of SEZ6ecto, which corresponds to the physiologically generated BACE1 cleavage fragment of SEZ6

(sSEZ6), rescued the glycosylation phenotype in SEZ6KO neurons, showed that the ectodomain is sufficient to promote GluK2/3 protein maturation. To exclude the possibility that sSEZ6—generated by BACE1 and found in the extracellular space—might

influence GluK2/3 protein maturation indirectly, we also tested whether BACE1 inhibition or deletion, which abolishes sSEZ6 secretion, has an effect. As expected, pharmacological BACE1 inhibition in primary cortical neurons and genetic BACE1 deletion in mouse

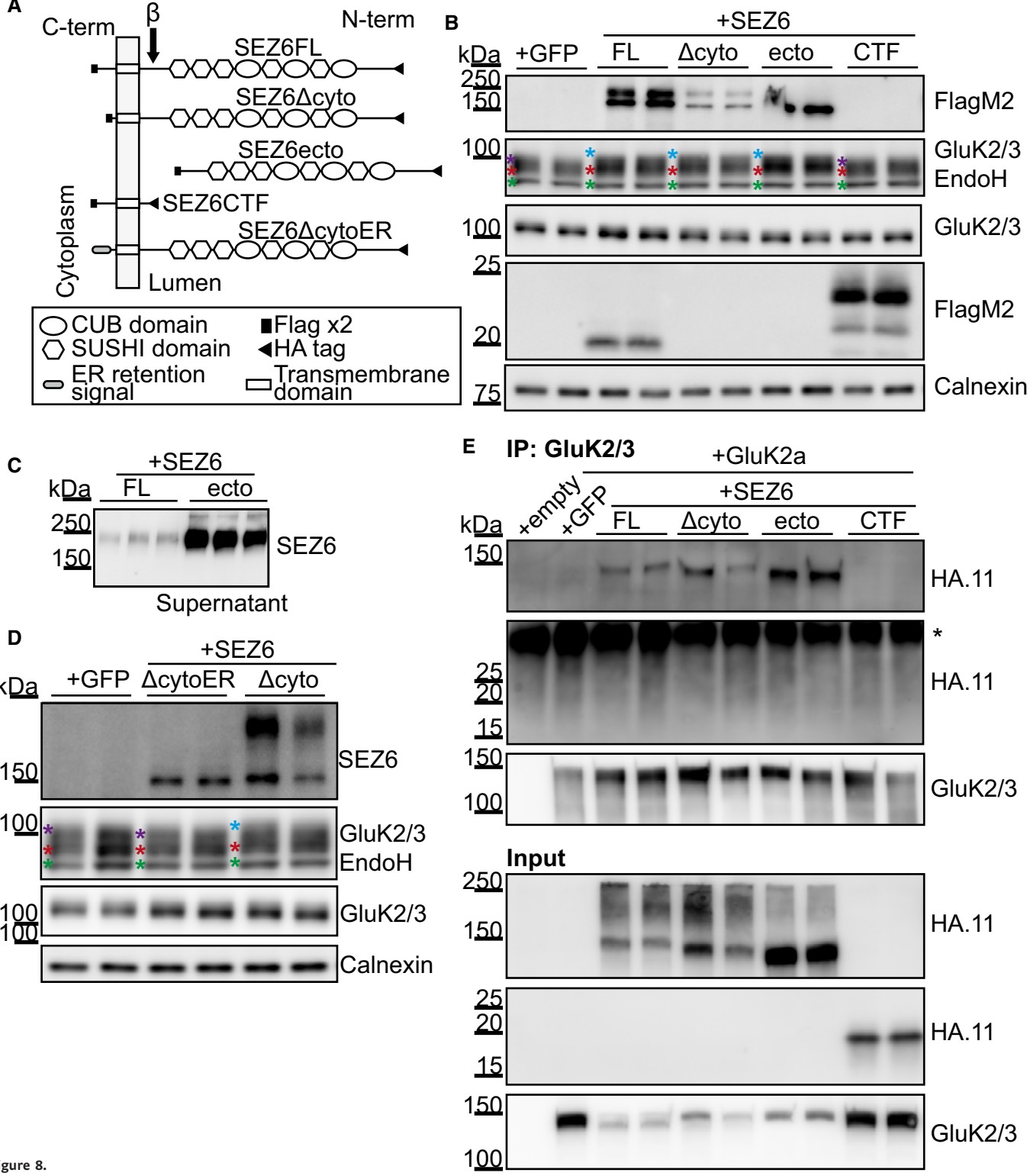

**Figure 8.**

◀

**Figure 8.  SEZ6 extracellular domain in the secretory pathway rescues GluK2/3 maturation in SEZ6KO neurons and binds to GluK2/3.**

A    Schematic representation of the constructs used: SEZ6 full length (SEZ6FL), SEZ6 lacking the C-terminal domain (SEZ6Δcyto), SEZ6 ectodomain generated by BACE1 cleavage (SEZ6ecto), SEZ6 C-terminal fragment generated by BACE1 cleavage (SEZ6CTF), and SEZ6 lacking the C-terminal domain but fused to an ER retention signal (SEZ6ΔcytoER).

B, C  SEZ6 constitutive KO neurons were transduced with SEZ6 constructs. SEZ6FL and SEZ6Δcyto were detectable as two bands corresponding to mature (higher) and immature (lower) forms of SEZ6. No difference in the apparent molecular weight was detected because the high percentage of the gel used (12%) and the small C-terminal deletion (39 amino acids). SEZ6ecto was present as the immature band in the neuronal lysates and as the mature, secreted band in the conditioned medium in panel C. SEZ6CTF (with HA and Flag-tags) was detected at the expected low molecular weight. Additionally, the C-terminal fragment generated from SEZ6FL by BACE1 cleavage was detected. Due to its lack of an HA-tag, it had a lower molecular weight than SEZ6CTF. The C-terminal fragment generated from SEZ6Δcyto upon BACE1 cleavage was not visible because of its low molecular weight. GluK2/3 glycosylation was rescued when SEZ6FL, SEZ6Δcyto, and SEZ6ecto were expressed in the neurons, but SEZ6CTF was not able to rescue the phenotype. Color coding for asterisks is the same as in Fig 4A. Representative blot of 3 independent experiments.

D    Sez6 constitutive KO neurons were transduced with SEZ6ΔcytoER and SEZ6Δcyto. SEZ6ΔcytoER presented only the band corresponding to the immature form of SEZ6, as expected for a protein retained in the ER. SEZ6Δcyto, but not SEZ6ΔcytoER, was able to rescue GluK2/3 glycosylation, showing that SEZ6 localized to the ER is not sufficient to rescue the GluK2/3 glycosylation ($n = 5$). Color coding for asterisks is the same as in Fig 4A.

E    HEK293T cells were co-transfected with a GluK2a plasmid and the SEZ6 constructs. The expression of SEZ6 constructs was analyzed in the total lysates ("Input") and revealed a similar expression of the different proteins. GluK2a was immunoprecipitated with GluK2/3 antibody, and the SEZ6 mutants were detected with the HA.11 antibody ("IP"). SEZ6FL, SEZ6Δcyto, and SEZ6ecto, but not SEZ6CTF were co-immunoprecipitated with GluK2a, suggesting that SEZ6 extracellular domain is necessary for GluK2a binding. Representative blot of 2 independent experiments. * indicates the light chain of the antibody used for immunoprecipitation.

Source data are available online for this figure.

brain homogenates neither altered GluK2/3 glycosylation nor cell surface levels of the GluK2/3 protein in primary neurons (Fig EV4). Thus, we conclude that the requirement of SEZ6 for GluK2/3 protein glycosylation and cell surface localization is independent of BACE1, suggesting that under physiological conditions, it is preferentially the full-length form of SEZ6 that controls GluK2 and/or GluK3 glycosylation and cell surface levels.

Our findings suggest that the regulation of GluK2 and/or three trafficking is mediated by the SEZ6 extracellular domain. To address the possibility of a direct binding of SEZ6 extracellular domain to GluK2 and/or GluK3, we first immunoprecipitated GluK2/3 from mouse brain extract and detected co-immunoprecipitated proteins by mass spectrometry. While the known GluK2/3 interactor Neto2 was detected (Zhang *et al*, 2009), SEZ6 was not detected (Table EV2). This suggests that the SEZ6 interaction with GluK2 and/or GluK3 is weaker or shorter-lived than the Neto2-GluK2/3 interaction and not stable enough to be detected in our extract conditions. Yet, it is possible that technical limitations prevented us from detecting an endogenous interaction of SEZ6 and GluK2. The successful demonstration of the interaction of two endogenous membrane proteins depends on the right choice of detergent. Thus, milder detergents than the NP40 used here may potentially allow the detection of endogenous SEZ6-GluK2 interactions.

Last, to test whether SEZ6 has the ability to interact with GluK2 and whether such interaction is mediated by the SEZ6 ectodomain, we co-expressed the GluK2a isoform with different SEZ6 mutants in HEK293 cells (Fig 8E). The SEZ6 mutants containing the extracellular domain (SEZ6FL, SEZ6ΔCyto, and SEZ6ecto) co-immunoprecipitated with GluK2a, but SEZ6CTF was not detectable (Fig 8E). These data demonstrate that SEZ6 is indeed able to form a complex with GluK2 and that the SEZ6 ectodomain is the site of interaction with GluK2.

## Discussion

Neurotransmitter receptors have fundamental functions in the nervous system, including for cellular responses to neurotransmission and synaptic plasticity. A fine-tuning of receptor activity is essential for adjusting neurotransmitter signal transduction to the required needs of a cell, organ, or whole organism. Here, we identified the CUB and CCP domain-containing protein SEZ6 as a new regulator of kainate receptors and demonstrate that SEZ6 controls the intracellular trafficking of kainate receptor subunits GluK2 and GluK3 to the cell surface, thereby controlling their glycosylation, cell surface localization, and ion channel activity in neurons (for a model, see Fig EV5). This reveals a novel molecular function for the transmembrane protein SEZ6, which is linked to schizophrenia and AD, but is also required for nervous system development, synaptic connectivity, and LTP (Gunnersen *et al*, 2007; Zhu *et al*, 2018).

The trafficking mechanism of action of SEZ6 on KAR function is surprising. Other CUB and CCP domain-containing proteins such as Neto1, Neto2, SOL-1, LEV-9, and LEV-10 act on iGluRs or AChRs (Nakayama & Hama, 2011). However, they bind stably to their receptors and act as auxiliary receptor subunits. For example, the auxiliary KAR subunits Neto1 and Neto2 bind GluK2 and specifically modulate key functional properties of GluK2, such as inducing slow channel kinetics and high agonist affinity, but also the synaptic localization of GluK2 (Zhang *et al*, 2009; Straub *et al*, 2011; Tang *et al*, 2011; Wyeth *et al*, 2014). In contrast to the Neto proteins, SEZ6 does not appear to fulfill the criteria established for auxiliary subunits (Copits & Swanson, 2012; Yan & Tomita, 2012) because our coimmunoprecipitations suggest that the SEZ6 interaction with GluK2 or GluK3 is more transient than the interaction between Neto2 and GluK2. Thus, we propose that SEZ6 is a trafficking factor for GluK2 and/or GluK3 rather than a stably interacting auxiliary subunit, thereby providing an additional layer of regulation of KARs beyond the auxiliary subunits. Interestingly, the SEZ6 homolog SEZ6L2 has recently been shown to bind AMPARs, but not KARs, in transfected cells (Yaguchi *et al*, 2017). Thus, the whole SEZ6 family consisting of SEZ6, SEZ6L, and SEZ6L2 may have important, but specific roles in iGluR trafficking.

Transmembrane proteins in the secretory pathway were long thought to traffic by default toward the plasma membrane, but it is now clear that several of them require specific transport helpers, such as iRhom2 for ADAM17, Cornichon for transforming growth factor, or ERGIC-53 for certain cathepsins and blood coagulation factors (Dancourt & Barlowe, 2010; Lichtenthaler, 2012). Through

its action on the secretory trafficking of GluK2, SEZ6 now joins this growing group of membrane protein transport helpers. The trafficking function of SEZ6 for GluK2 and/or GluK3 is supported by several findings: Firstly, co-expression of SEZ6 with GluK2 promoted trafficking of GluK2 through the secretory pathway, as measured with the RUSH system in HEK293 cells and in primary neurons. In addition, the use of an ER-retained SEZ6 construct revealed that SEZ6 specifically controls the post-ER trafficking of GluK2 and/or GluK3. Conversely, loss of SEZ6 reduced neuronal surface levels of GluK2 and/or GluK3 in a post-transcriptional manner with only a minor effect on total protein levels of GluK2/3 in neurons and no effect in mouse brains. Importantly, loss of SEZ6 did not affect cell surface levels of other detected iGluRs and most other transmembrane proteins, revealing a specific effect on GluK2 and/or GluK3 surface localization. Finally, as required for a trafficking factor, SEZ6 bound GluK2 in co-immunoprecipitation experiments and this interaction occurred through its CUB domain-containing ectodomain. Potentially, SEZ6 affects interactions of GluK2 and/or 3 with other proteins, such as protein kinase C and PDZ ligand interactions, that control correct positioning of GluK2 at the plasma membrane (Evans et al, 2017).

Importantly, in the absence of SEZ6, surface levels of GluK2 and GluK3 were reduced by about 50%, but were not completely abolished, indicating that GluK2 and GluK3 surface transport can—at least partly—happen in the absence of SEZ6. Other proteins may allow the remaining cell surface transport in the absence of SEZ6. Alternatively, it is possible that SEZ6 is required for GluK2 and GluK3 transport in only some, but not all neurons in the brain. In fact, while GluK2/GluK3 and SEZ6 are coexpressed in some areas of the brain, such as the CA1 region of the hippocampus and mid-cortical layers of the cortex (Foster et al, 1981; Kim et al, 2002; Straub et al, 2011), GluK2 and GluK3 are also expressed in other brain areas, such as the CA3 region of the hippocampus, where SEZ6 is barely expressed (Kim et al, 2002; Pigoni et al, 2016; Watanabe-Iida et al, 2016).

Loss of SEZ6 not only reduced trafficking and surface localization of GluK2 and GluK3, but additionally prevented GluK2 and/or GluK3 from carrying the post-translational modification HNK-1, which is a covalently attached sulfated sugar chain consisting of three different sugar molecules, glucuronic acid, galactose, and N-acetylglucosamine (HSO3-3GlcAβ1-3Galβ1-4GlcNAc) (Chou et al, 1986). The unique glycostructure that distinguishes HNK-1 from other glycostructures is the terminal sulfated glucuronic acid (Appendix Fig S2). HNK-1 has a functional role in cell migration, adhesion, and recognition and is an autoantigen in peripheral demyelinating neuropathy (Morise et al, 2017). Addition of the HNK-1 epitope to proteins occurs in the trans-cisternae of the Golgi or within the trans-Golgi network (Kizuka & Oka, 2012), but the mechanisms or sequence features that control HNK-1 addition to proteins remain unknown. With our finding that the immunoblot signal intensity for HNK-1 was unaltered in SEZ6KO mouse brain extracts, we conclude that SEZ6 is mechanistically not generally required for HNK-1 modification of proteins, but instead specifically enables HNK-1 modification of GluK2 and/or GluK3. We therefore consider that SEZ6 is either required for transporting GluK2 and 3 to the cellular site of HNK-1 modification or that it facilitates HNK-1 modification of GluK2 and/or GluK3. Whether the lack of the HNK-1 modification in SEZ6KO neurons in turn also contributes to the

reduced GluK2/3 surface levels or is independent of the reduced surface levels is not yet clear. However, HNK-1 is generally able to alter protein trafficking, as shown for another iGluR subunit, the AMPAR subunit GluA2. In that case, loss of HNK-1 increased GluA2 endocytosis and reduced cell surface levels of GluA2 in hippocampal neurons (Morita et al, 2009).

Our study shows that SEZ6 is required for normal secretory pathway trafficking and cell surface localization of GluK2/3, so that the amplitude of kainate-evoked currents in the ex vivo system of acute hippocampal slices from SEZ6KO mice was reduced. Additionally, the lack of HNK-1 on GluK2/3 in SEZ6KO neurons may also contribute to the reduced kainate-evoked current. While it has not yet been investigated whether and how the loss of HNK-1 alters GluK2 and/or GluK3 function, the opposite experiment was done (Vernon et al, 2017). Co-expression of GluK2a or GluK3a with the two HNK-1 synthesizing enzymes in HEK293 cells enabled HNK-1 modification on both GluK2a and GluK3a and resulted in glutamate-evoked currents with slower desensitization kinetics. The mean peak amplitude for the current was not altered for GluK2a but increased threefold for GluK3a (Vernon et al, 2017). Thus, it appears possible that SEZ6 acts on GluK2/3 function through both mechanisms, i.e., through controlling surface levels and independently through HNK-1 modification of GluK2/3.

SEZ6 is linked to neurological and psychiatric diseases, but the underlying molecular mechanisms are little understood. Genetic variants of SEZ6 are linked to childhood-onset schizophrenia (Thr229-Thr231del) (Ambalavanan et al, 2015), intellectual disability (Arg657Gln) (Gilissen et al, 2014), and AD (Arg615His) (Paracchini et al, 2018). These mutations are localized within the SEZ6 ectodomain that interacts with GluK2 and/or GluK3. With our newly established SEZ6 function for surface levels and HNK-1 modification of GluK2 and/or GluK3, it appears possible that these mutations act as loss-of-function mutations and cause reduced KAR activity, which may result in altered synaptic plasticity and LTP (Bortolotto et al, 1999; Contractor et al, 2001; Pinheiro et al, 2007; Sherwood et al, 2012). Interestingly, besides SEZ6 mutations, also changes in HNK-1 metabolism may contribute to psychiatric diseases. In fact, single nucleotide polymorphisms or chromosome breakpoint translocation sites close to HNK-1 synthesizing enzymes were genetically linked to schizophrenia (Jeffries et al, 2003; Kahler et al, 2011).

Not only full-length, but also the soluble SEZ6 ectodomain (sSEZ6) has been linked to disease. While increased sSEZ6 levels in CSF were reported in bipolar, depressive and schizophrenic patients (Maccarrone et al, 2013), and inflammatory pain conditions (Roitman et al, 2019), the opposite was seen in AD, where reduced levels of the sSEZ6 were reported in CSF (Khoonsari et al, 2016). However, it remains unclear whether these changes directly contribute to disease pathogenesis or are merely a consequence of the disease process. sSEZ6 is released from full-length SEZ6 through the action of the protease BACE1 (also known as β-secretase) (Pigoni et al, 2016), which is a key drug target in AD as it catalyzes the first step in the generation of the pathogenic Aβ peptide (Vassar et al, 2014). Our study revealed that the knock-out of BACE1 and the use of BACE inhibitors did not affect glycosylation or cell surface trafficking of GluK2/3. Thus, the action of full-length SEZ6 on GluK2/3 function was independent of BACE1. This is good news for the clinical development of BACE1-targeted inhibitors, as they are not

expected to cause side effects by affecting GluK2 and/or GluK3 function. Yet, the BACE1 cleavage products of SEZ6, sSEZ6 or the C-terminal fragment (SEZ6CTF), may have physiological functions other than full-length SEZ6 and such functions may still be affected by the clinically tested BACE inhibitors. In fact, pharmacological BACE1 inhibition in mice reduced LTP and dendritic spine density in a SEZ6-dependent manner (Zhu *et al*, 2018). While the underlying molecular mechanisms still need to be elucidated, these experiments imply BACE1 cleavage products in controlling LTP and spine density. Finding different functions for full-length vs. soluble SEZ6 is reminiscent of other single-span transmembrane proteins, such as the death receptor DR6 (Colombo *et al*, 2018), the neuronal cell adhesion protein NrCAM (Brummer *et al*, 2019), and the B-cell maturation antigen (Laurent *et al*, 2015), where the full-length protein and cleavage products have different physiological functions.

In summary, our study identifies the neuronal protein SEZ6 as a novel trafficking protein of KARs that controls activity, localization, and glycosylation of the KAR subunits GluK2 and GluK3. This reveals for the first time a molecular function for the transmembrane protein SEZ6, which has a fundamental role in the brain, such as for nervous system development, synaptic connectivity, and long-term potentiation (Gunnersen *et al*, 2007; Zhu *et al*, 2018). Given the genetic link of SEZ6 to psychiatric and neurologic diseases, the new function and mechanism of action for SEZ6 are also of major relevance for understanding these devastating diseases.

# Materials and Methods

## Materials

The following antibodies were used as follows: monoclonal SEZ6 (Pigoni *et al*, 2016), polyclonal SEZ6 (Gunnersen *et al*, 2007), pAb SEZ6L2 (R&D Systems, AF4916), pAb SEZ6L (R&D Systems, AF4804), GluR6/7 (04-921, Millipore, GluK2 and GluK3 antibodies commercially available are not able to discriminate between these two subunits due to their high homology), NMDAR2b (D15B3, Cell Signaling), anti-GluR2 (MAB397, Millipore), 3D5 (kindly provided by Robert Vassar), calnexin (Enzo, Stressgen, Farmingdale, NY, USA, ADI-SPA-860), β-tubulin (T8578, Sigma), β-actin (Sigma, A5316), PSD 95 (2507, Cell Signaling), LDLR (R&D Systems, AF2255), NCAM-1 (R&D Systems, AF6070), rat mAb Flag M2 (F1804, Sigma), 5F8 anti-Red (Chromotek), anti-HA.11 (MMS-101P, Covance), HRP-coupled anti-mouse and anti-rabbit secondary (DAKO), and HRP-coupled anti-goat, anti-rat, and anti-sheep (Santa Cruz).

The following reagents and media were used as follows: neurobasal medium, HBSS and B27 (Invitrogen), C3 (β-secretase inhibitor IV; Calbiochem, 565788, final concentration 2 μM), DMEM (Gibco), and FBS (Thermo Fisher Scientific).

## Mouse strains

The following mice were used in this study: wild-type (WT) C57BL/6NCrl (Charles River); BACE1$^{-/-}$ (Jackson Laboratory, strain B6.129-Bace1tm1Pcw/J, BACE1KO); SEZ6$^{-/-}$ (SEZ6KO) (Gunnersen

*et al*, 2007); SEZ6 flox/flox (Gunnersen *et al*, 2007); SEZ6 family triple knock-out (TKO) mice lacking SEZ6, SEZ6L, and SEZ6L2 (Miyazaki *et al*, 2006); SEZ6L$^{-/-}$ [SEZ6LKO, bred from SEZ6 family TKO (Miyazaki *et al*, 2006)]; and SEZ6L2$^{-/-}$ [SEZ6L2 KO, bred from SEZ6 family TKO (Miyazaki *et al*, 2006)]. All mice were on a C57BL/6 background and were maintained on a 12/12-h light–dark cycle with food and water *ad libitum.*

## Summary mouse brains and cortical neurons used in the paper

SEZ6 flox/flox cortical neurons were used in Figs 1, 2, 4A, 5A, and EV1, Appendix Fig S2, Fig EV4A and C. SEZ6 flox/flox hippocampal neurons were used in Fig 7. SEZ6KO cortical neurons were used in Fig 8B–D, and Appendix Fig S3. SEZ6KO adult brains were used in Figs 3, 4B, 4E, 4F, 5B, and EV2B. BACE1KO brains Fig EV4B.

## Lentiviral plasmids

pFU HA-SLIC-Flagx2-mmSEZ6FL was generated synthetizing full-length *Mus musculus* SEZ6, transcript variant 1 (UniProt Q7TSK2-1) in pFU vector, where the insert replaced the original GFP in the pFUGW vector (Lois *et al*, 2002). The signal peptide of SEZ6 was maintained, followed by a short spacer (SLIC), and an HA tag (YPYDVPDYA). A double FLAG tag (DYKDDDDK) was cloned to the C-terminus of the protein. The different SEZ6 mutants were similarly expressed in the pFU vector, and GFP or empty pFU were used as control. In particular, pFU HA-SLIC-Flagx2-mmSEZ6Δcyto was generated removing the last 39 amino acids at the C-terminus of the protein. The pFU HA-SLIC-Flagx2-mmSEZ6ecto and pFU HA-SLIC-Flagx2-mmSEZ6 CTF containing the SEZ6 ectodomain and the SEZ6 C-terminal fragment, respectively, were generated according to the BACE1 cleavage site located between leucine906 and aspartate907 (Pigoni *et al*, 2016). For the generation of the SEZ6 ER retention mutant, the pFU HA-SLIC-Flagx2-mmSEZ6Δcyto was used as template and the Flagx2 was substituted with the WBP1 ER retention signal (KKLETFKKTN) (Shikano & Li, 2003). Lentiviruses encoding Cre recombinase or an inactivated Cre mutant (Y324F) as the negative control were generated by cloning the Cre inserts into pF2U, which is largely identical to pFU, but lacks a 500 bp stretch of nucleotides between the WPRE element and the 3′ LTRsin, thus allowing larger inserts to be cloned into the vector. The inactivated Cre construct was generated using the quick-change mutagenesis strategy with the primers pair: fw: 5′-TGAACTTCATCAGAAACCTG-GACTCTGAGAC-3′; rev: 5′-CTGATGAAGTTCATCACAATGTTCA-CATTG-3′. The two helper plasmids psPAX2 and pcDNA3.1(-)-VSVG were used for the packaging of the lentiviral particles.

## Isolation of primary cortical neurons and lentiviral infection

Neuronal cultures were derived from SEZ6 flox/flox and SEZ6KO mice at E15.5/E16.5. Brains from fetuses were prepared and digested with papain (Sigma) for 30 min. Tissue was triturated, and cortical neurons were separated by sequential passage of the cell suspension through plastic pipettes. Cells were centrifuged for 3 min at 800 *g* and resuspended in seeding medium (DMEM containing 10% FBS). The cell number was determined, and neurons were seeded at a density of 1.5 million cells per milliliter in

Poly-D-Lysin (Sigma-Aldrich)-coated plates. In general, neurons were infected with lentiviruses expressing GFP, CRE, or the different constructs described above, at day 2 *in vitro* (DIV) with the exception of the rescue experiment in Fig 8 (infection was done at DIV 0 in order to maximize the rescue effect). After 5 days *in vitro* (DIV), neurons were washed with PBS, medium was replaced with fresh neurobasal media supplemented with B27, and the experiments were carried out at DIV 7.

### SUSPECS labeling of proteins for mass spectrometry analysis

Surface labeling and processing of the samples were performed as previously described (Herber *et al*, 2018). Briefly, 40 million primary cortical SEZ6 flox/flox neurons were infected at two DIV with CRE (SEZ6KO neurons) or GFP (control) virus. At five DIV, medium was replaced with fresh neuronal culture medium supplemented with 50 μM Ac4-ManNAz. At DIV7, biotinylation of glycoproteins at the cell surface was performed via bioorthogonal click chemistry applying 100 μM DBCO-PEG12-biotin (Click-chemistry tools) for 2 h at 4°C. Cells were lysed in STET-lysis buffer (150 mM NaCl, 50 mM Tris (pH 7.5), 2 mM EDTA, 1% Triton X-100) with protease inhibitor, centrifuged and lysates were filtered through a 0.45 μm syringe filter (Millipore). Cell lysates of three control (GFP) and three SEZ6KO (CRE) were loaded on a polyprep chromatography column (Bio-Rad) containing 300 μl of high-capacity streptavidin agarose beads (Thermofisher) and washed with 10 ml 2% SDS in PBS to remove non-specifically bound proteins. Streptavidin beads were dried completely, and proteins were eluted from the beads by boiling 5 min at 95°C in 150 μl Laemmli buffer supplemented with 8 M urea and 3 mM biotin. Samples were separated with 10% SDS–polyacrylamide gel electrophoresis (PAGE) and stained with 0.025% (*w/v*) Coomassie Brilliant Blue in 10% acetic acid. The gel was destained in 10% acetic acid, and each lane was cut into 14 horizontal slices (=14 fractions) at equal height and subjected to tryptic in-gel digestion. In Appendix Fig S3, SUSPECS was similarly used to label and biotinylate the glycoproteins at the cell surface. Proteins were enriched with streptavidin agarose beads, eluted, and loaded on the gel for GluK2 and/or three detection via immunoblot.

### In gel-digestion and peptide purification

In-gel digestion and peptide purification were performed as previously described (Shevchenko *et al*, 2006). Briefly, proteins residing in the gel were denatured with 10 mM dithiothreitol (DTT) in 100 mM ammonium bicarbonate (ABC), reduced with 55 mM iodoacetamide (IAA) in 100 mM ABC, and proteolytic digestion was performed at 37°C overnight using 150 ng trypsin per fraction. 40% acetonitrile (ACN) supplemented with 0.1% formic acid was used to extract the peptides. Peptides were dried by vacuum centrifugation and reconstituted in 0.1% formic acid for proteomic analysis.

### LC-MS/MS analysis

Each gel fraction was analyzed on an Easy nLC-1000 (Thermo Scientific, USA), which was coupled online via a nanoelectrospray source (Thermo Scientific, USA) equipped with a PRSO-V1 column oven (Sonation, Germany) to a Velos Pro Orbitrap Mass Spectrometer (Thermo). Peptides were separated on a self-packed C18 column (300 mm × 75 μm, ReproSil-Pur 120 C18-AQ, 2.4 μm, Dr. Maisch, Germany) with a binary gradient of water (A) and acetonitrile (B) containing 0.1% formic acid (0 min, 2% B; 3:30 min 5% B; 48:30 min, 25% B; 59:30, 35% B; 64:30, 60% B) at a flow rate of 250 nl/min and column temperature of 50°C. Full MS spectra were acquired in profile mode at a resolution of 30,000 covering a *m/z* range of 300–2,000. The ten most intense peptide ions per full MS scan were chosen for collision-induced dissociation (CID) within in the ion trap (isolation width: 2 *m/z*; normalized collision energy: 35%; activation *q*: 0.25; activation time: 10 ms). A dynamic exclusion of 40 s was applied for peptide fragmentation. Two technical replicates were acquired per sample.

### LC-MS/MS data analysis and statistical evaluation

Database search and label-free quantification were performed with the software MaxQuant (version 1.4.1.2, maxquant.org) (Cox & Mann, 2008). Trypsin was defined as protease (cleavage specificity: C-terminal of K and R). Carbamidomethylation of cysteines was defined as fixed modification. Oxidation of methionines and acetylation of protein N-termini were defined as variable modifications. Two missed cleavages were allowed for peptide identification. The first search option was enabled to recalibrate precursor masses using the default values. The data were searched against a mouse database including isoforms (UniProt, download: May 16, 2014; 51,389 entries). The false discovery rate (FDR) was adjusted to less than 1% for both, peptides and proteins. Common contaminants such as bovine proteins of fetal calf serum and human keratins were excluded. Label-free quantification (LFQ) intensity values were used for relative quantification. At least two ratio counts of razor and unique peptides were required for protein quantification. The LFQ intensities of the technical replicates were averaged, and LFQ ratios (SEZ6KO/WT) were calculated separately for each biological replicate. The LFQ ratios were $\log_2$ transformed, and a one-sample *t*-test ($\mu 0 = 0$) was applied to identify proteins with a significant abundance difference at the cell surface proteome.

### Cell lysate preparation

In general, supernatants and lysates from neurons were collected at DIV7 as previously described (Dislich *et al*, 2015). In order to detect better the difference in the GluK2/3 glycosylation, Triton lysis buffer [150 mM NaCl, 50 mM Tris–HCl pH 7.5, 1% Triton X-100, and protease inhibitor cocktail (Roche)] was used. BCA assay (Uptima Interchim, UP95425) was used to quantify protein concentrations, and 10–15 μg of total neuronal lysate was used for Western Blot analysis.

### Brain homogenization

Brains were isolated from SEZ6KO mice and WT at different age. All brains were homogenized in 20 mM HEPES pH 7.4, 150 mM NaCl, 0.5% NP-40, 2 mM EDTA, 10% Glycerol and incubated on ice for at least 1 h. Samples were then centrifuged at maximum speed for 5 min in order to remove the membranes. BCA assay (Uptima

Interchim, UP95425) was used to quantify protein concentrations, and 15–20 µg of total protein was used for Western blot analysis.

## Brain fractionation

Brains were isolated from P7 BACE1 KO mice and WT littermates kindly provided by Prof. Jochen Herms. Samples were processed as previously described (Kuhn et al, 2012), and the concentration of the membrane proteins was quantified with a BCA assay (Uptima Interchim, UP95425). 15–20 µg of total protein was used for Western blot analysis.

## Western blot analysis

Samples were boiled for 5 min at 95°C in Laemmli buffer and separated on 8% or 12% SDS–polyacrylamide handcast gels or 4–12% MOPS gradient gels (GenScript). PVDF membranes (Millipore) were incubated with primary antibody for 1–2 h at room temperature or at 4°C overnight. After incubation with secondary antibody at room temperature for 1 h, membranes were developed with ECL prime (GE Healthcare, RPN2232V1).

## Deglycosylation assay

10–12 µg of neuronal lysate or brain homogenate was treated with endoglycosidase H (Endo H, New England Biolabs, P0702) or Peptide-N-Glycosidase F (PNGase F, New England Biolabs, P0704) according to the manufacturer's protocol. Afterwards, the samples were separated on 8% SDS–polyacrylamide gel and Western blotting was performed.

## Surface biotinylation

Neurons were biotinylated at DIV7 with EZ-Link™ Sulfo-NHS-Biotin (ThermoFisher, 21217) according to manufacturer's protocol. Quenching was done with ammonium chloride (50 mM) and BSA (1%) in PBS and lysis with SDS lysis buffer (50 mM Tris–HCl pH 8, 150 mM NaCl, 2 mM EDTA, 1% SDS). To dilute the samples, RIPA buffer (10 mM Tris–HCl pH 8, 150 mM NaCl, 2 mM EDTA, 1% Triton, 0.1% sodium deoxycholate, 0.1% SDS) was used. After sonication, protein concentrations were quantified and 80 µg of total lysate was incubated with 25 µl of High Capacity Streptavidin Agarose Resin (ThermoFisher, 20361). Samples were incubated rotating 2 h at room temperature or overnight at 4°C. Beads were washed with RIPA buffer, and bound proteins were eluted in Laemmli buffer supplemented with 3 mM biotin by boiling at 95°C. Eluted proteins were separated on 8% SDS–polyacrylamide gel, and Western blot analysis was performed.

## RNA extraction and RT–qPCR

RNA was extracted from DIV7 neurons using the RNeasy Mini Kit (QIAGEN), and reverse transcription was performed using the High-Capacity cDNA Reverse Transcription Kit (ThermoFisher) according to manufacturer's protocol. RT–qPCR was carried out using the StepOne-Plus real-time PCR system (Life Technologies) and power Sybr Green master mix (Applied Biosystems). Reaction volumes of 20 µl with the following specific primers (0.5 µM) were used (Rangel et al, 2009):

| Gene | Forward primer 5′–3′ | Reverse primer 5′–3′ |
|---|---|---|
| GluA1 | CTCGCCCTTGTCGTACCAC | GTCCGCCCTGAGAAATCCAG |
| GluA2 | GTGTCGCCCATCGAAAGTG | AGTAGGCATACTTCCCTTTGGAT |
| GluK2 | ATCGGATATTCGCAAGGAACC | CCATAGGGCCAGATTCCACA |
| GluK3 | AGGTCCTAATGTCACTGACTCTC | GCCATAAAGGGTCCTATCAGAC |
| GluN2B | GCCATGAACGAGACTGACCC | GCTTCCTGGTCCGTGTCATC |
| β-Actin | CCCAGAGCAAGAGAGG | GTCCAGACGCAGGAT |
| GAPDH | AGGTCGGTGTGAACGGATTTG | TGTAGACCATGTAGTTGAGGTCA |

Amplification conditions consisted of 15″ denaturation at 95°C, 1 min of annealing and elongation at 60°C for 40 cycles. The results were normalized by the expression levels of *gapdh and β-actin*. Data were analyzed by StepOne Software (Applied Biosystems) following the $2^{-\Delta\Delta CT}$ method (Livak & Schmittgen, 2001).

## Analysis of GluK2 editing

cDNA extract from DIV 7 neurons (see session RNA extraction and RT–qPCR) was used as template to amplify the M2 region of GluK2. The following primers were used: GluK2 5′-GGTATAATCGAC ACCCTTGCAACC-3′, GluK2 5′-TGACTCCATTAAGAAAGCATAATC GGA-3′. BbvI (New England Biolabs) digestion was performed according to manufacturer's protocol and was used to determine the level of GluK2 RNA editing (Bernard et al, 1999). The digested product was run on 2% agarose gel, and bands were quantified using NIH ImageJ. To determine the level of editing, was used the formula (intensity of 376 [edited]/intensity of [376 (edited) + 269 (unedited)]) × 100, according to Evans et al (2017). The band at 76 bp was used to determine equal loading.

## Preparation of acute hippocampal slices

Wild-type and SEZ6$^{-/-}$ mice on a C57BL/6 background at postnatal day P15 were used in the experiments. After the mice were deeply anesthetized with $CO_2$ and decapitated, the brain was immediately removed and immersed in ice-cold slicing solution containing (in mM) 24.7 glucose, 2.48 KCl, 65.47 NaCl, 25.98 NaHCO$_3$, 105 sucrose, 0.5 CaCl$_2$, 7 MgCl$_2$, 1.25 NaH$_2$PO$_4$, and 1.7 ascorbic acid (Fluka, Switzerland).The pH value was adjusted with to 7.4 with HCl and stabilized by bubbling with carbogen which contained 95% $O_2$ and 5% $CO_2$, and the osmolality was 290–300 mOsm. 300 µm horizontal hippocampal slices were cut in the slicing solution by the use of a vibratome (VT1200S; Leica, Germany). Brain slices were kept in a recovering solution which contained (in mM) 2 CaCl$_2$, 12.5 glucose, 2.5 KCl, 2 MgCl$_2$, 119 NaCl, 26 NaHCO$_3$, 1.25 NaH$_2$PO$_4$, 2 thiourea (Sigma, Germany), 5 Na-ascorbate (Sigma), 3 Na-pyruvate (Sigma), and 1 glutathione monoethyl ester (Santa Cruz Biotechnology, USA) at room temperature for at least 1 h before the experiment. The pH value of the recovering solution was adjusted to 7.4 with HCl and constantly bubbled with carbogen, and the osmolality was 290 mOsm.

## Electrophysiological recordings

After resting in the recovery solution for at least 1 h, individual hippocampal slices were transferred to the recording chamber,

which was constantly perfused at a flow rate of 3 ml/min with artificial cerebrospinal fluid (ACSF) containing (in mM) 2 $CaCl_2$, 20 glucose, 4.5 KCl, 1 $MgCl_2$, 125 NaCl, 26 $NaHCO_3$, and 1.25 $NaH_2PO_4$ and gassed with 95% $O_2$ and 5% $CO_2$ to ensure oxygen saturation and to maintain a pH value of 7.4. 30 μM D-AP5 (Abcam and Tocris, USA), 20 μM GYKI53655 (Tocris), and 10 μM bicuculline (Enzo, USA) were added to the ACSF to block NMDAR-, AMPAR-, and $GABA_A$R-mediated synaptic transmission. Somatic whole-cell recordings from CA1 pyramidal neurons were performed with a borosilicate glass pipette with the resistance of *ca.* 7 MΩ filled with internal solution which contained (in mM) 148 K-gluconate, 10 HEPES, 10 NaCl, 0.5 $MgCl_2$, 4 Mg-ATP, and 0.4 $Na_3$-GTP. The pH value of internal solution was adjusted to 7.3 with KOH. Voltage-clamp measurements were carried out using an EPC9/2 patch-clamp amplifier (HEKA, Germany). The membrane potential was held at −70 mV in voltage-clamp mode without liquid junction potential adjustment. Data acquisition and the generation of stimulation protocols were applied by the use of PULSE software (HEKA). Data were collected at 10 kHz and Bessel-filtered at 2.9 kHz and analyzed through Igor 5 software (Wavemetrics, USA).

### Synaptosomes

Synaptosomes were purified from full mouse brain (without olfactory bulbs and cerebellum) according to Carlin *et al* (1980). Brains were homogenized in 0.32 M sucrose, 1 mM $NaHCO_3$, 1 mM $MgCl_2$, and 0.5 mM $CaCl_2$ and diluted at a final concentration of 10% *w/v*. Samples were centrifuged at 1,400 *g* for 10 min at 4°C. The supernatants (S1) were transferred to a fresh tube and centrifuged at 13,800 *g* for 10 min at 4°C. The supernatants (S2) were discarded and pellets (P2) resuspended in 0.32 M sucrose, 1 mM $NaHCO_3$. Sucrose gradient was set up by slowly over-laying increasing sucrose concentration solutions (0.85 M sucrose, 1M sucrose, and 1.2 M sucrose in water). P2 suspensions were loaded carefully on the sucrose layers, and tubes were centrifuged in an ultracentrifuge with swing out rotor at 82,500 *g* for 2 h at 4°C (max acceleration, but slow deceleration). Synaptosome fractions were collected between layers 1.2 and 1 M sucrose, and centrifuged at 13,800 *g* for 15 min at 4°C. Sucrose was completely removed and pellets (P3) solved in SDT buffer (2% SDS, 100 mM Tris, 50 mM DTT).

### Quantitative analysis of the EndoH-digested GluK2/3 bands

Immunoblotting images were quantified along the *y*-axis by ImageJ software, where "Lane analysis" function used individual lanes of the clockwise rotated image. The X–Y coordinates of the resulting plots were exported. The coordinates were refined first by subtracting the background signal, and then by determining the start and the end points of the signal coming from the immunoblot band. For quartile analysis, the individual lane values were first corrected for area differences. This allowed the comparison of the signal distribution of the lanes independent of their signal intensity. Average signal intensity within a quartile was calculated and represented in the table in Fig EV2A. Student's *t*-test applied to the WT-SEZ6KO comparison, while analysis of the rescue experiments in Fig 8 was done using one-way-ANOVA test with Dunnett's multiple comparison correction.

### Lectin chip microarray (LecChip)

For lectin chip microarray analysis (LecChip), primary neuronal cultures from WT ($n = 3$) and SEZ6KO mice ($n = 4$) were first lysed in STET-Buffer (50 mM Tris–HCl, pH 7.5, 150 mM NaCl, 2 mM EDTA, 1% Triton X-100, protease inhibitor mix 1:500) and brought to identical protein concentrations (measurement was performed by BCA reagent, Uptima Interchim, UP95425) in TBS to obtain a 0.05 mg/ml solution. LecChip (GlycoTechnica Ltd.) was washed three times with Probing Solution (provided by the manufacturer), and lysates (1 μg/ml) were added to the wells in Probing Solution. Samples were incubated overnight at 18°C. After incubation, an excess blocker glycoprotein was added to the chip and incubated for 30 min. The Blocking solution was then discarded, and the chip was washed three times with 0.1% TBST. Anti-GluR6/7 antibody (04-921, Millipore) in blocking solution in TBS was then applied to the chip and incubated for 1 h. After three washes with 0.1% TBST, AF555-labeled goat anti-rabbit (Life Technologies) solution in 0.1% TBST was added and incubated for 30 min. Subsequently, the LecChip was washed with TBS and double-distilled water 30 min each. The LecChip was scanned with InnoScan 710 Microarray scanner (Innopsys), and results were analyzed with CLIQS Array Professional software (TotalLab). Overall, 45 lectin intensities were measured in each sample. An average Pearson correlation coefficient of 0.98 and above was calculated for all samples. Lectin intensities were considered, and discoveries were determined using the Original FDR method of Benjamini and Hochberg, with Q = 2%.

### Immunoprecipitation and HNK-1 detection

Protein G agarose beads preconjugated with GluR6/7 (1.5 μl of antibody and 40 μl of beads per sample) or NCAM-1 antibody (12.5 μl of antibody and 40 μl of beads per sample) were incubated with 200 ug of total brain homogenates O/N at 4°C. After washing, proteins were eluted in Laemmli Buffer, boiled at 95°C for 5 min, and loaded on a 4–12% MOPS gradient gel (GenScript) or 8% gel. Detection was done with HNK-1 1C10 supernatant (Developmental Studies Hybridoma Bank). As control for the specificity of the HNK-1 antibody, the proteins pulled down from two WT samples were digested with PNGase F before elution to remove HNK-1 epitope.

### RUSH Cargo Sorting Assay using confocal microscopy in HEK293T cells

RUSH Cargo Sorting Assay was performed as described previously (Deng *et al*, 2018). SBP-GFP-GluK2 construct was kindly provided by Jeremy Henley (Evans *et al*, 2017). HEK293T cells (ATCC, CRL-3216) were cultured on sterile glass slides coated with fibronectin (10 μg/1 ml DPBS, overnight, 4°C Millipore Sigma, F0895) in 6-well plates. For transfection, cells were seeded at density $2 \times 10^5$ in each well of 6-well plates. After 24 h, cells were transfected with 1 μg of SBP-GFP-GluK2 and 0.5 μg of pFU HA-SLIC-Flagx2-mmSEZ6FL or pFU HA-SLIC-mmSEZ6ΔcytoER or pIRESneo3-SBP-LyzC-RFP (where LyzC-RFP was cloned into Addgene plasmid 65264: pIRESneo3-Str-KDEL_ST-SBP-EGFP) (Boncompain *et al*, 2012), respectively, using Lipofectamine 2000 (Invitrogen, 116680-019). After 48 h, cells were incubated with 40 μM d-Biotin (SUPELCO) in DMEM for 0, 20, and 40 min. As a control, cells without d-Biotin were monitored to

confirm retention of the reporter in the ER. After washing with 1× PBS, cells were fixed with 4% paraformaldehyde in PBS for 10 min and prepared for immunofluorescence microscopy. To visualize ER and Golgi, cells were incubated over night with either Calnexin (mouse, BD 610524, dilution 1:500), GM130 (mouse, BD 610457, dilution 1:500), or TGN46 (sheep, Biorad AHT500GT, dilution 1:500) antibodies, respectively. To visualize HA-tagged Sez6 proteins, cells were incubated overnight at 4°C with anti-HA.11 antibody (mouse, BioLegend 901533, dilution 1:1,000) followed by an incubation with an secondary donkey anti-mouse antibody labeled with Alexa 594 (Life Technologies, A21203, dilution 1:1,000) for 1 h at room temperature. Hoechst staining of nuclear DNA was performed for 10 min at room temperature. Sample images were acquired using a DeltaVision Elite workstation (Applied Precision) based on an inverted OLYMPUS IX-70 microscope using a 100×, 1.4 NA oil immersion lens. To image the complete cell, 10–18 stacks in Z-direction with a step/size of 0.2 μm were recorded of each field of view. Images were captured with a sCMOS camera (CoolSnap HQ; Photometrics) and deconvolved with softWoRx (v.6.0) software using the iterative-constrained algorithm and the measured point spread function. For the analysis with ImageJ/Fiji, only cells showing transport of the reporter from the ER to Golgi after Biotin incubation were considered, while cells showing ER signal after Biotin addition were excluded (1% of the cell pool, 99% of the cells, the protein nicely moves from the ER to the Golgi after the addition of biotin). For quantification of these occurring cytosolic vesicles, we empirically measured the sizes of objects between 4 and 20 pixels using the Analyze Particles function in ImageJ, which detects vesicular structures but omits larger structures such as the Golgi. While small-fragmented and isolated Golgi structures could be detected in error, such structures are rare. Furthermore, only vesicles of cells expressing the HA-tagged as well as the RUSH-construct were counted. The Fiji macro *count_fixed_vesicles_V1.3* (M. Pakdel) including the *Particle Analyzer* plug-in by Fiji was used to determine the number of vesicles. In Fig EV3D, the RUSH system was similarly used with the difference that SBP-mCherry-GluK2 and inactive CRE were used as fluorescently tagged cargo and negative control, respectively.

### GluK2 RUSH assay in primary hippocampal neurons

Hippocampal neurons were isolated from E16.5 WT or SEZ6 flox/flox embryos and cultured on poly-D-lysine-coated glass coverslips in 12-well plates at a density of 120,000 cells per well. SEZ6 flox/flox neurons were infected at DIV0 with lentiviruses encoding Cre recombinase (pF2U-Cre), or an inactivated Cre mutant (Y324F) as the negative control. Neurons were transfected at DIV7 with the SBP-mCherry-GluK2 construct kindly provided by Jeremy Henley (Evans *et al*, 2017) using Lipofectamine 2000 following the manufacturer's instructions. At DIV10, neurons were washed twice with PBS and treated with 40 μM D-biotin for 20 or 40 min together with the anti-SBP antibody (1:500, Millipore) diluted in HBSS. As a control, WT neurons were incubated with anti-SBP antibody without biotin for 40 min. After the anti-SBP live-labeling, neurons were washed twice with PBS, fixed in 4% PFA, and immunostained for Cre recombinase (D7L7L, Cell Signaling) followed by fluorescently labeled secondary antibodies to SBP (mouse Alexa488) and Cre (Rabbit Alexa647). Quantification of GluK2 surface/total ratio

(SBP/mCherry) was performed as described previously (Evans *et al*, 2017). Briefly, images for total GluK2 (mCherry) and surface GluK2 (SBP) were acquired using a Leica SP5 confocal microscope. ImageJ/Fuji was used to measure the mean fluorescence intensity of multiple region-of-interests (ROIs) including the neuronal soma and primary neurites. For one cell, the same ROIs for mCherry image and anti-SBP image were measured and the surface/total ratio (SBP/mCherry) was then calculated. At least thirteen neurons were analyzed per condition, and cells were obtained from at least two independent experiments.

### Inhibitor treatment of neurons

The BACE inhibitor C3 (β-secretase inhibitor IV; Calbiochem, 565788, final concentration 2 μM) was applied to neurons at DIV 5 supplemented to fresh medium. Treatment was prolonged for 48 h and lysates collected at DIV 7.

### Co-immunoprecipitation

HEK293T was transfected with 1 μg of pcDNA3.6 mycGluR6a(Q) plasmid kindly provided by Prof. Christophe Mulle (Jaskolski *et al*, 2005) and 0.5 μg of different SEZ6 and SEZ6L2 constructs (see previous description). 72 h later, cells were lysated in 20 mM HEPES pH 7.4, 150 mM NaCl, 0.5% NP-40, 2 mM EDTA, and 10% Glycerol and lysates were incubated with protein G agarose beads preconjugated with GluR6/7 antibody (10 μls of antibody and 300 μl of beads were used for 14 samples) O/N at 4°C. After washing, proteins were eluted in Laemmli Buffer, boiled at 95°C for 5 min, and loaded on a 4–12% MOPS gradient gel (GenScript). Detection was done with GluR6/7 (04-921, Millipore) and anti-HA.11 (MMS-101P, Covance) antibodies. For the Co-immunoprecipitation followed by mass spectrometry analysis, 50 μl of protein G agarose beads per sample was conjugated with 5 μl of GluR6/7 antibody or 600 μl SEZ6 14E5 hybridoma supernatant o/n, at 4°C, with rotation. Conjugated beads were washed three times with PBS, and 50 μl of the bis(sulfosuccinimidyl)suberate (BS3) solution (2.5 mM BS3 in PBS) was added and incubated 1 h, RT, rotating. BS3 crosslinked PGS beads were then washed in different solutions: three times with 50 μl of 100 mM glycine pH 2.8, two times with PBS supplemented with 1% Igepal CA-630 (*v/v*), and finally one time with PBS. After washing, crosslinked beads were incubated with 2 mg of brain homogenate in the HEPES NP-40 buffer O/N, at 4°C on rotator. Samples were washed three times with HEPES NP-40 buffer and eluted in 30 μl of 8% formic acid for 10 min, RT.

### Preparation of CoIP samples for mass spectrometry analysis

The protein concentrations of the co-IP samples were estimated using the Pierce 660 nm assay supplemented with the ionic detergent compatibility reagent (Thermo Fisher Scientific, USA). A protein amount of 20 μg was subjected to proteolytic digestion using the filter-aided sample preparation (FASP) protocol with 30 kDa Vivacon filters (Sartorius, Germany) as previously described (Wisniewski *et al*, 2009). Peptides were purified with self-packed C18 stop and go extraction (STAGE) tips as previously described (Rappsilber *et al*, 2003).

## MS analysis of Co-IP samples

Co-IP samples were analyzed on an Easy nLC 1200 nanoHPLC (Thermo Scientific) which was coupled online via a Nanospray Flex Ion Source (Thermo Scientific, USA) equipped with a PRSO-V1 column oven (Sonation, Germany) to a Q-Exactive HF mass spectrometer (Thermo Scientific, USA). An amount of 1.3 μg of peptides per sample was separated on an in-house packed C18 column (30 cm × 75 μm ID, ReproSil-Pur 120 C18-AQ, 1.9 μm, Dr. Maisch GmbH, Germany) using a binary gradient of water (A) and acetonitrile (B) supplemented with 0.1% formic acid (0 min, 2% B; 3:30 min, 5% B; 137:30 min, 25% B; 168:30 min, 35% B; 182:30 min, 60% B) at 50°C column temperature. A data-dependent acquisition method was used. Full MS scans were acquired at a resolution of 120,000 ($m/z$ range: 300–1,400, AGC target: 3E+6). The 10 most intense peptide ions per full MS scan were selected for peptide fragmentation (resolution: 15,000, isolation width: 1.6 $m/z$, AGC target: 1E+5, NCE: 26%). A dynamic exclusion of 120 s was used for peptide fragmentation.

The raw data were analyzed with the software Maxquant (maxquant.org, Max-Planck Institute Munich) version 1.5.5.1 (Cox *et al*, 2014). The MS data were searched against a reference fasta database of Mus musculus from UniProt (download: March 09, 2017, 16,851 entries). Trypsin was defined as protease. Two missed cleavages were allowed for the database search. The option first search was used to recalibrate the peptide masses within a window of 20 ppm. For the main search, peptide and peptide fragment mass tolerances were set to 4.5 and 20 ppm, respectively. Carbamidomethylation of cysteine was defined as static modification. Acetylation of the protein N-term as well as oxidation of methionine was set as variable modifications. The false discovery rate for both peptides and proteins was adjusted to less than 1%. The "match between runs" option was enabled with a matching window of 2 min. Label-free quantification (LFQ) of proteins required at least two ratio counts of razor or unique peptides. Only razor and unique peptides were used for quantification.

## Ethics approval and consent to participate

All animal procedures were performed in accordance with either the European Communities Council Directive (86/609/EEC) or Australian Code of Practice for the Care and Use of Animals for Scientific Purposes. Animal protocols were approved by the Ludwigs-Maximilians-University Munich and the government of Upper Bavaria, or alternatively the Anatomy & Neuroscience, Pathology, Pharmacology, and Physiology Animal Ethics Committee of the University of Melbourne, Australia.

## Statistics

In general, data were analyzed using the Mann–Whitney test and considered significant when the *P*-value was lower than 0.05. In general, at least six replicates from two biological replicates or four independent biological replicates were considered, but the exact number of replicates is specified for each experiment in the figure legend. In Fig 5, discoveries were determined using the FDR method of Benjamini and Hochberg, with Q = 2%, and at least three mouse

brains per genotype were used. Mass spectrometry data were analyzed as described in the previous sections.

## Data availability

This study includes no data deposited in external repositories.

**Expanded View** for this article is available online.

## Acknowledgements

We thank Katrin Moschke, Anna Berghofer, Claudia Ihbe, and Marek-Jan Czyz for their technical assistance. We thank Hiroshi Takeshima for providing triple SEZ6 KO mice. This work was supported by the Deutsche Forschungsgemeinschaft (German Research Foundation) under Germany's Excellence Strategy within the framework of the Munich Cluster for Systems Neurology (EXC 2145 SyNergy—ID 390857198) and the research unit FOR2290, by the Centers of Excellence in Neurodegeneration, the BMBF through grant CLINSPECT-M, the JPND project PMG-AD, and the Bayerisches Hochschulzentrum für Mittel-, Ost- und Südosteuropa (BAYHOST). M.A.B. and J.H. are supported by the UK Dementia Research Institute which receives its funding from DRI Ltd, funded by the Medical Research Council, Alzheimer's Society and Alzheimer Research UK (UKDRI-1010). M.A.B. is also funded by a UKRI Future Leaders Fellowship (MR/S017003/1) and a grant from the BrightFocus Foundation (A2019112S). J.v.B was funded by the National Institute of General Medical Sciences of the United States National Institutes of Health, award number GM134083-01. J.M.G. is funded by the Australian National Health and Medical Research Council (GNT10008046, GNT1058672, GNT1140050). TK has been supported by Parkinson Fonds Deutschland, Hilde Ulrichs Stiftung, Friede Springer Stiftung, Deutsche Stiftung für Neurologie, Lüneburg Heritage.

## Author contributions

MP and SFL conceptualized the study; H-EH, JH, JRN, MDS, JT, RK, PG, MLT, BR, BB, ALHE, JVB, and TK investigated the study; MP, H-EH, SAM, GG, and GR involved in formal analysis; SAM, GG, and MW curated the data; GG involved in visualization; P-HK, CM, and GR contributed to methodology; JMG provided resources; SFL supervised the study, MP and SFL wrote—original draft; MP, JVB, CM, MW, GR, MAB, JG, TK, SFL wrote—review & editing.

## Conflict of interest

The authors declare that they have no conflict of interest.

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
