## [Review Process File · The EMBO Journal]

Seizure protein 6 controls glycosylation and trafficking of kainate receptor subunits GluK2 and GluK3

Martina Pigoni, Hung-En Hsia, Jana Hartmann, Jasenka Njavro, Merav Shmueli, Stephan Müller, Gökhan Güner, Johanna Tüshaus, Peer-Hendrik Kuhn, Rohit Kumar, Pan Gao, Mai Ly Tran, Bulat Ramazanov, Birgit Blank, Agnes Hipgrave Ederveen, Julia Von Blume, Christophe Mülle, Jenny Gunnensen, Manfred Wuhrer, Gerhard Rammes, Marc Aurel Busche, Thomas Koeglsperger, and Stefan Lichtenthaler

DOI: 10.15252/emboj.2019103457

Corresponding author(s): Stefan Lichtenthaler (stefan.lichtenthaler@dzne.de)

Review Timeline:

Submission Date:	14th Sep 19
Editorial Decision:	15th Oct 19
Revision Received:	8th Apr 20
Editorial Decision:	11th May 20
Revision Received:	18th May 20
Accepted:	20th May 20

Editor: Karin Dumstrei

Transaction Report:

Dear Stefan,

Thank you for submitting your manuscript to The EMBO Journal. Your study has now been seen by three referees and their comments are provided below.

As you can see from these comments, the referees find the analysis interesting and are overall supportive of publication. They raise a number of different concerns that I would like to ask you to address in a revised version.

Let me know if we need to discuss any issues in further detail - I am happy to do so.

When preparing your letter of response to the referees' comments, please bear in mind that this will form part of the Review Process File, and will therefore be available online to the community. For more details on our Transparent Editorial Process, please visit our website:

<https://www.embopress.org/page/journal/14602075/authorguide#transparentprocess>

Thank you for the opportunity to consider your work for publication. I look forward to your revision.

with best wishes

Karin

Karin Dumstrei, PhD
Senior Editor
The EMBO Journal

Further information is available in our Guide For Authors:

The revision must be submitted online within 90 days; please click on the link below to submit the revision online before 13th Jan 2020.

Link Not Available

Referee #1:

The report describes a new role for SEZ6 in trafficking of GluK2/3 kainate receptor subunits in in vitro and in vivo contexts. SEZ6 is a CUB and Sushi domain containing type I integral membrane protein that is important for the development of the mammalian nervous. It's also a substrate for BACE1 cleavage to shed an ectodomain. The connection between SEZ6 and the KARs was discovered through an unbiased proteomic comparison of wildtype and SEZ6-knockdown using a clever N-glycosylation based assay. The reduction in GluK2/3 was then validated in biochemistry and physiological assays. A strong case is made for SEZ6 acting as a forward trafficking chaperone that promotes complex oligosaccharide processing in the Golgi/trans-Golgi compartment, based on HNK-1 modification of the subunit proteins. Finally, the authors examined the domain(s) in SEZ6 that were required for the KAR interaction. In all, the authors did a thorough job of supporting their primary conclusion; the experiments are rigorous and span reduced systems to the mouse brain.

Major concerns: none

Minor concerns

The SUSPECS approach could be explained in the Results to a greater extent. For example, it's not clear what output this approach yields without digging into the Methods, how reproducible the

results are in the independent assays, and how glycosylated proteins were differentiated from non-glycosylated proteins. It's not a common technique, having just been described last year, and for that reason would benefit from a bit more elaboration.

The data from the SUSPECS/mass spectrometry analysis should be included as supplemental information. Also, there appear to be a fair number of other proteins that are down-regulated (and up-regulated) in the knockout cortical neurons. Is there a central functional theme amongst those proteins in the most impacted groups?

A few points in the Discussion:

- To what extent (and where) do SEZ6 and GluK2/3 expression overlap in the CNS?
- The fact that Neto1/2 alter KAR trafficking and synaptic localization is no longer a matter of uncertainty, as the authors state on p. 13. See Copits et al., 2011; Wyeth et al., 2014; Orav et al., 2017 (as a few examples).

In the bibliography, the Herber et al. entry is incomplete.

Referee #2:

This manuscript describes a molecular function for the Seizure protein 6 (SEZ6), as a facilitator of the intracellular trafficking and maturation of kainate receptors (KARs) in the brain. SEZ6 is particularly expressed in neurons and it is genetically associated to psychiatric and neurodegenerative disorders. However, there is very little information on this protein. From earlier studies on the SEZ6 knockout, it was proposed that this protein is involved in neuronal development and synaptic function, but no information on molecular or cellular mechanisms was available. In this study, the authors have found that SEZ6 is required for the GluK2/3 subunits of KARs to acquire a mature glycosylation pattern, associated to the secretory trafficking and surface expression of the receptor. As a functional assay, the authors also observe that KAR-mediated currents in CA1 neurons are reduced in the absence of SEZ6. The authors also map the ectodomain of SEZ6 as the interacting region with GluK2/3, and identify the human natural killer-1 (HNK-1) as the glycan modification that is defective in the absence of SEZ6.

These findings are mechanistically novel and physiologically relevant, as they help to understand the neuronal function of this protein. Nevertheless, some of the mechanistic aspects of this study are not sufficiently explored, and the physiological relevance of some of the assays may be enhanced. These are my specific comments:

1. This study starts with a proteomic approach developed by the authors, to identify surface proteins containing complex glycosylation. By comparing primary neuronal cultures with or without SEZ6 genetic deletion, they determine that only a handful of proteins require SEZ6 for their glycosylation and surface expression. The GluK2 and GluK3 subunits of KARs are among the proteins most clearly affected by SEZ6 deletion. This result is very clear, as the validation by surface biotinylation in Fig. 2A and Suppl. Fig. 1. Nevertheless, the authors also acknowledge that global expression of GluK2/3 is also affected by the lack of SEZ6 (Fig. 4). Therefore, surface-to-total ratios of GluK2/3 should be explicitly quantified and plotted, to properly evaluate specific effects on surface trafficking.
2. As a functional assay, the authors show that whole-cell kainate-induced currents are reduced in CA1 pyramidal neurons from acute hippocampal slices of the SEZ6 knockout mouse (Fig. 3). While this is important information, it is hard to evaluate the functional relevance of these currents. For

example, postsynaptic KARs are known to contribute little to the EPSC, but they regulate neuronal excitability by inhibiting postspike afterhyperpolarization. Also, presynaptic KARs at mossy-fibers are well established modulators of neurotransmitter release. A more physiological form of KAR function should be assessed. At this point, some sentences in the manuscript are clearly overstated (Discussion: "Here we identified... SEZ6 as a new regulator of neurotransmission"; Abstract: "SEZ6... thereby fine-tuning neurotransmission...").

3. The experiment of synchronized forward trafficking in Fig. 6 is one of the critical ones in this study, but it is very poorly presented.

a. What is the green channel in panel b? Just because of the colored fonts, one would assume that the green signal comes from the co-expressed protein (together with the mCherry-GluK2), but this should be explicitly declared. In fact, in the Methods section, only the name of the primary antibodies is described (not their species), so the reader cannot know with which antibody (and fluorophore) these antibodies were coupled.

b. The choice of iCRE as a control is rather odd. This is a nuclear protein, and this is why the green signal appears concentrated in the nucleus of the cell only for this data set. An ER- or Golgi-resident protein may have been a better control.

c. What is the criterion to identify "Golgi-derived vesicles", as stated in the legend for Fig. 6B? Again, co-expression (or co-labeling) with a Golgi marker may give this information. Otherwise, white arrowheads in panel b are just pointing to vesicles.

d. In the Methods section, it is stated that "only cells showing transport of the reporter from the ER to Golgi after biotin incubation were considered, while cells showing ER signal after biotin addition were excluded". What does this mean? In the sample figures in panel b, there is plenty of red signal not present in vesicles. How can you determine that this is not ER? Overall, the quality of these images seems too low to make conclusions about compartment identities.

e. Do the authors extract (or can extract) any conclusion from the extent of colocalization between SBP-mCherry-GluK2 and SEZ6 (or SEZ6L2)? Potentially, this type of images may give valuable information on the co-transport of SEZ6 with GluK2, but again, images of much better quality would be required.

4. The authors argue that HEK293 cells are used for the trafficking assay to avoid competition with endogenous protein. Still, given the specializations of neurotransmitter receptor trafficking in neurons, a trafficking assay of labeled GluK2/3 on wt or SEZ6 knockout neurons would be much more convincing. In fact, taking into account my comment #2, I consider that a more physiological assay in neurons, either for receptor trafficking or for the modulation of synaptic function, would significantly strengthen the mechanistic interpretation of the role of SEZ6 in the regulation of KAR function.

5. How were SEZ6 constructs expressed in neurons for Fig. 7 (panels B-D)? This is a very important experiment, as it evaluates the effect of SEZ6 mutants on the maturation of endogenous GluK2/3 in neurons. However, there is no information in the figure legend or the Methods section (it is just stated that neurons were "transduced"). In order to see changes in glycosylation of endogenous GluK2/3, the fraction of transduced cells had to be close to 100%. No viral vector is mentioned in the text, and plasmid transfection on neurons is typically far from 100% efficient.

Minor point.

6. I am not sure it is justified to state that SEZ6 cannot be considered an auxiliary protein for KARs because their co-immunoprecipitation from brain extracts was negative (Discussion). Many factors could affect the outcome of an immunoprecipitation, and it is always risky to extract conclusions from negative results.

Referee #3:

Summary: The authors report the results of proteomics from membranes of WT and SEZ6 KO mice. Given the relevance of SEZ6 in various disease the results of this study is important and of interest in the field. However, the major concerns are the lack of quantitative evidence for the conclusion drawn and the unconvincing nature of many of the immunoblots shown.

Major concerns:

Figure 3: Auxiliary factors such as SOL-1 and NETO alter biophysical properties of iGluRs. The authors should test if there are alterations in such properties in neurons and heterologous cells with co-expression of SEZ6.

Page 8: The statement, "Total levels of the GluK2/3 bands were not changed and this was also seen for a control protein, the GluA2 subunit of AMPA receptors (Fig. 4B and D)" is misleading because total levels were significantly decreased in neuronal lysates (Figure 4C) but not brain lysates (Figure 4D). Furthermore, quantification for GluA2 as control should be included in the graphs in Figures 4C and 4D.

Figure 4E: There appears to be a reduction in the level of SEZ6 in the SEZ6L2KO; thus, one might expect changes in the glycosylation level of GluK2/3, at least to a lesser extent, as shown in SEZ6 KO lysates (Fig. 4A). The blots shown in Figure 4E should be quantified to determine if there is a significant decrease in SEZ6, and if so, provide an explanation of the lack of glycosylation changes in GluK2/3.

Figure 4F: The conclusion that, "demonstrating that the effect of SEZ6 on maturation of GluK2 and/or GluK3 is independent of the developmental stage and that no compensation occurs through other proteins during adulthood," is confusing as it would be expected that GluK2/3 maturation deficits would be dependent on SEZ6 level. In addition, the glycosylation pattern observed at E16.5 is very different from later ages and this pattern is dependent on SEZ6 level during developmental stage. This paragraph should be rewritten to accurately reflect the data shown and conclusions that can be drawn.

Page 9: The authors state that, "So far, our experiments revealed that SEZ6 is required for normal cell surface localization and signaling as well as for correct glycosylation of GluK2 and/or GluK3 with the HNK-1 epitope." It is not clear to what 'signaling' refers, do the authors mean 'ion channel conductance' as shown in Figure 3? Please clarify this term. The term signaling is used throughout the discussion and should be clarified here as well.

Figure 6: The images shown are not particularly convincing. What is the green immunofluorescence? The authors state that, "SBP-mCherry-GluK2 was retained in the ER (Fig. 6B)" without use of any ER marker to validate this statement at the 0 min time point. Similarly, no Golgi marker is used to confirm forward trafficking to this organelle at 20 min. Furthermore, the authors state that cytoplasmic vesicles appear at the 40 min time point but there are also vesicular structures pointed out and quantified in the 20 min time point. Also, no explanation as to how 'vesicles' are quantified is provided in the methods.

Figure 7: It is difficult to judge 'rescue' of GluK2/3 maturation based on the stated 'reappearance' of the uppermost band. The authors should perform biotinylation studies as in Figure 2A which allow for quantification of the extent of rescue. The results of the glycosylation state could then be interpreted in that context. It is surprising that SEZ6 Δ Cyto was expressed at significantly lower intensity on the blot and yet could 'rescue' the upper mature band for GluK2/3 and should be

explained. Quantification of the 'rescue' might help to understand the lack of relationship between expression level of the constructs and surface expression of GluK2/3. Furthermore, the blot shown in Figure 7D is not convincing to demonstrate that SEZ6/delta-cyto can 'rescue' the mature glycosylation compared with SEZ6/delta-cytoER, even with the apparent longer running of the gel as evidenced by the increase area between the upper and lower bands of SEZ6 in Figure 7D compared with Figure 7B. Furthermore, the IP results shown in Figure 7E require explanation as to the significance of the HA.11 signal just above the 25 kDa molecular weight marker. It also appears that co-expression of the SEZ6 constructs, except for the CTF, negatively impacts the level of GluK2/3 in HEK293 cells.

Minor concerns:

Introduction page 5: Neto binds Kainate receptors not AChR or AMPAR as stated.

Page 7: The conclusion that, "Taken together, these experiments suggest that SEZ6 controls cell surface levels of GluK2/3 at the post-translational level," should be updated to include reference to the modest effect on total GluK2/3 levels (15% reduction). For example, "Taken together, these experiments suggest that SEZ6 primarily controls cell surface levels of GluK2/3 at the post-translational level; however, a modest effect on total levels, indicating reduced biosynthesis, was also observed."

Page 11: The statement that, "suggesting that under physiological conditions it is preferentially the full-length form of SEZ6 that controls SEZ6 glycosylation and cell surface levels". - this should be 'controls GluK2/3 glycosylation and cell surface levels.'

Figure 4A: The arrows in the panel at right are distracting and should be replaced with asterisks as in the EndoH portion of the panel. The indication 'partially EndoH resistant' is confusing, to what is this arrow referring? There is a purple asterisk that seems to refer to a blue band, which is probably supposed to also be purple. Furthermore, the asterisks in the SEZ6 blot (upper left) are not explained.

Figure 4B: The glycosylation patterns in Figures 4A and 4B appear different; thus, it would be useful to include a zoomed-in panel as shown for Figure 4A.

Figures 5-7: It is not stated how many replicates of the glycosylation or immunoprecipitation experiments were performed in the legend or methods.

We thank the Referees for their encouraging and constructive reviews. A point-by-point response to each of their comments is provided below.

Reviewer #1

The report describes a new role for SEZ6 in trafficking of GluK2/3 kainate receptor subunits in in vitro and in vivo contexts.

...

In all, the authors did a thorough job of supporting their primary conclusion; the experiments are rigorous and span reduced systems to the mouse brain.

Major concerns: none

Minor concerns

The SUSPECS approach could be explained in the Results to a greater extent. For example, it's not clear what output this approach yields without digging into the Methods, how reproducible the results are in the independent assays, and how glycosylated proteins were differentiated from non-glycosylated proteins. It's not a common technique, having just been described last year, and for that reason would benefit from a bit more elaboration.

According to the suggestion, we described the method in more details. The added text on page 5 of the results is: "In this method, neurons are metabolically labeled with a chemically modified, azido group-containing sugar (N-azidomannosamine, ManNAz). The azido group is later used for click chemistry-mediated biotinylation of glycoproteins. After uptake into neurons, ManNAz is converted to azido-sialic acid, which gets incorporated into the glycan chains of newly synthesized cellular proteins, when they traffic through the trans-Golgi network. After two days of metabolic labeling the living neurons were biotinylated with dibenzylcyclooctyne (DBCO)-biotin, which covalently reacts with the azido group of the glycoproteins residing at the surface and late in the secretory pathway. Biotinylated proteins were enriched with streptavidin agarose and analyzed by mass spectrometry using label-free protein quantification."

We also described how glycosylated and non-glycosylated proteins were differentiated. We focused on proteins being annotated in the Uniprot database as being glycosylated. The new sentence on page 5 is now: "3209 proteins were detected in 3 out of 3 experiments by the SUSPECS analysis and 571 were glycosylated, according to Uniprot annotation (Fig. 1B and Supplementary Table 1)".

Additionally, we now provide a figure on the reproducibility of the method (new suppl. Fig. 1) and describe this in the text on page 5: "The sample preparation workflow showed generally little variation between samples, as indicated with correlation coefficients of larger than 0.94 between different samples (Supplementary Fig. 1)".

The data from the SUSPECS/mass spectrometry analysis should be included as supplemental information. Also, there appear to be a fair number of other proteins that are down-regulated (and up-regulated) in the knockout cortical neurons. Is there a central functional theme amongst those proteins in the most impacted groups?

As suggested, we added the mass spectrometric results as new suppl. Table 1.

Moreover, we added a new paragraph in the results' section on page 6 and briefly described the other changed proteins. There was no obvious central functional theme among these proteins, and a pathway analysis was not feasible given the low number of changed proteins. The next text reads as: "Other integral membrane proteins significantly reduced on the cell surface of SEZ6KO neurons were Ceramide synthase 2 (-50%), which is involved in sphingolipid and myelin biosynthesis (Imgrund et al., 2009) and Syntaxin-12 (Stx12, also known as Stx13 in humans; -50%), which is an endosomal trafficking protein contributing to internalization and recycling of various cell surface receptors (Wang and Tang, 2006). Integral membrane proteins with less pronounced surface changes were the neuronal cell adhesion protein L1cam, Paralemmin-1 (Palm), the glial cell line-derived neurotrophic factor (GDNF) family receptor alpha-1 (Gfra1), the Golgi-localized protein Kiaa1467, which is also known as FAM234B and the glycosylation enzyme N-acetylgalactosaminyltransferase 1 (Galnt1) (Supplementary Table 2). The ER-localized lysophospholipid acyltransferase 2 (Mboat2) was the only integral membrane protein with increased abundance in the SEZ6KO. The changed membrane proteins did not have obvious commonalities, and a pathway analysis was not feasible given the low total number of changed proteins. Thus, we focused on the two KAR subunits GluK2 and GluK3, which showed the strongest reduction".

A few points in the Discussion:

-To what extent (and where) do SEZ6 and GluK2/3 expression overlap in the CNS?

As suggested, we included a short paragraph into the discussion on page 15: "In fact, while GluK2/GluK3 and SEZ6 are coexpressed in some areas of the brain, such as the CA1 region of the hippocampus and mid-cortical layers of the cortex (Foster et al., 1981, Kim et al., 2002, Straub et al., 2011), GluK2 and GluK3 are also expressed in other brain areas, such as the CA3 region of the hippocampus, where SEZ6 is barely expressed (Kim et al., 2002, Pignoni et al., 2016, Watanabe-Iida et al., 2016)."

-The fact that Neto1/2 alter KAR trafficking and synaptic localization is no longer a matter of uncertainty, as the authors state on p. 13. See Copits et al., 2011; Wyeth et al., 2014; Orav et al., 2017 (as a few examples).

Thanks very much for the references. We included the Wyeth reference, as it specifically deals with GluK2 – the protein that we investigate. The new sentences on page 14 of the discussion are: "For example, the auxiliary KAR subunits Neto1 and Neto2 bind GluK2 and specifically modulate key functional properties of GluK2, such as inducing slow channel kinetics and high agonist affinity, but also the synaptic localization of GluK2 (Straub et al., 2011, Tang et al., 2011, Wyeth et al., 2014, Zhang et al., 2009)".

In the bibliography, the Herber et al. entry is incomplete.

The entry has been corrected.

Referee #2:

This manuscript describes a molecular function for the Seizure protein 6 (SEZ6), as a facilitator of the intracellular trafficking and maturation of kainate

receptors (KARs) in the brain. SEZ6 is particularly expressed in neurons and it is genetically associated to psychiatric and neurodegenerative disorders. However, there is very little information on this protein. From earlier studies on the SEZ6 knockout, it was proposed that this protein is involved in neuronal development and synaptic function, but no information on molecular or cellular mechanisms was available. In this study, the authors have found that SEZ6 is required for the GluK2/3 subunits of KARs to acquire a mature glycosylation pattern, associated to the secretory trafficking and surface expression of the receptor. As a functional assay, the authors also observe that KAR-mediated currents in CA1 neurons are reduced in the absence of SEZ6. The authors also map the ectodomain of SEZ6 as the interacting region with GluK2/3, and identify the human natural killer-1 (HNK-1) as the glycan modification that is defective in the absence of SEZ6.

These findings are mechanistically novel and physiologically relevant, as they help to understand the neuronal function of this protein. Nevertheless, some of the mechanistic aspects of this study are not sufficiently explored, and the physiological relevance of some of the assays may be enhanced. These are my specific comments:

1. This study starts with a proteomic approach developed by the authors, to identify surface proteins containing complex glycosylation. By comparing primary neuronal cultures with or without SEZ6 genetic deletion, they determine that only a handful of proteins require SEZ6 for their glycosylation and surface expression. The GluK2 and GluK3 subunits of KARs are among the proteins most clearly affected by SEZ6 deletion. This result is very clear, as the validation by surface biotinylation in Fig. 2A and Suppl. Fig. 1. Nevertheless, the authors also acknowledge that global expression of GluK2/3 is also affected by the lack of SEZ6 (Fig. 4). Therefore, surface-to-total ratios of GluK2/3 should be explicitly quantified and plotted, to properly evaluate specific effects on surface trafficking.

As suggested, we quantified the surface-to-total ratios of GluK2/3 in the absence and presence of SEZ6. The results are shown in new suppl. figure 2B and demonstrate that the surface/total ratio is still significantly reduced in the absence of SEZ6. This trafficking function of SEZ6 is in line with the new GluK2 RUSH experiment in primary neurons (new Fig. 7), where SEZ6 promoted surface transport of GluK2.

The new text in the results' section on page 6 is: "Interestingly, in addition to the 50% surface reduction, total GluK2/3 protein levels in neuronal SEZ6KO lysates were also reduced, but only by about 15% (see later Fig. 4C). Because the surface levels were more strongly reduced than the total levels, the surface/total ratio of GluK2/3 was significantly reduced in the SEZ6KO neurons compared to the WT control (Supplementary Fig. 2B)."

2. As a functional assay, the authors show that whole-cell kainate-induced currents are reduced in CA1 pyramidal neurons from acute hippocampal slices of the SEZ6 knockout mouse (Fig. 3). While this is important information, it is hard to evaluate the functional relevance of these currents. For example, postsynaptic KARs are known to contribute little to the EPSC, but they regulate neuronal excitability by inhibiting postspike afterhyperpolarization. Also, presynaptic KARs at mossy-fibers are well established modulators of neurotransmitter release. A more physiological form of KAR function should be assessed. At this point, some sentences in the manuscript are clearly

overstated (Discussion: "Here we identified... SEZ6 as a new regulator of neurotransmission"; Abstract: "SEZ6... thereby fine-tuning neurotransmission...").

This important reviewer comment is similar to comment #4 below by the same reviewer, where the reviewer combines this comment #2 and the comment #4 below and asks to include a **'more physiological assay in neurons, either for receptor trafficking or for the modulation of synaptic function'**. We chose to include a new figure with a physiological trafficking assay in neurons for GluK2 trafficking (described below at point #4), because of the following reason. Functional studies with KARs are typically carried out in the CA3 region of the hippocampus. To test whether lack of SEZ6 affects GluK function in CA3, as measured with any of the typically used assays, SEZ6 would need to be coexpressed with GluK in this brain area. However, while GluK2 and GluK3 are clearly expressed in CA3, SEZ6 is not expressed in CA3, as we showed previously (Pigoni et al. Mol Neurodegeneration 2016). Thus, it is well possible that SEZ6 does not affect trafficking of all GluK2 or GluK3-containing KARs, but controls their function in a region-dependent manner. In contrast to CA3, SEZ6 is ubiquitously expressed in CA1, where also GluK2/3 are expressed. This is the reason why we focused on CA1 in the first version of our manuscript and recorded kainate-evoked currents in acute hippocampal slices in WT and SEZ6KO mice in CA1 (Fig. 3). We made this point clearer in the results, where the new sentence on page 7 is: "Because SEZ6 is not expressed in the CA3 region of the hippocampus (Pigoni et al., 2016), we focused on the CA1 region, where both SEZ6 and GluK2 are expressed (Bureau et al., 1999, Kim et al., 2002, Lein et al., 2007)."

Additionally, we changed the suggested sentences. The new sentence in the discussion is now: "Here, we identified the CUB and CCP domain-containing protein SEZ6 as a new regulator of kainate receptors and demonstrate that SEZ6 controls the intracellular trafficking of kainate receptor subunits GluK2 and GluK3 to the cell surface, thereby controlling their glycosylation, cell surface localization and ion channel activity in neurons (for a model see Supplementary Fig. 8)."

The last sentence of the abstract was also rewritten and is now: "Taken together, SEZ6 acts as a new trafficking factor for GluK2/3. This novel function may help to better understand the role of SEZ6 in neurologic and psychiatric diseases."

3. The experiment of synchronized forward trafficking in Fig. 6 is one of the critical ones in this study, but it is very poorly presented.

a. What is the green channel in panel b? Just because of the colored fonts, one would assume that the green signal comes from the co-expressed protein (together with the mCherry-GluK2), but this should be explicitly declared. In fact, in the Methods section, only the name of the primary antibodies is described (not their species), so the reader cannot know with which antibody (and fluorophore) these antibodies were coupled.

Thank you for pointing this out. We have taken out the previous version of Fig. 6 and provide a new Figure 6 with new experiments and more details, including a new Fig. 7 and a new suppl. Fig. 5. Also, we split the channels, and labeled the images correspondingly (**Figure 6B and 6C**). We provide additional information in the figure legends (page 39) and inserted details on antibodies and fluorophores in the updated

version of the Materials and Methods section on pages 27-28. Within the results' section, the new experiments are described from page 9, bottom, to page 11.

b. The choice of iCRE as a control is rather odd. This is a nuclear protein, and this is why the green signal appears concentrated in the nucleus of the cell only for this data set. An ER- or Golgi-resident protein may have been a better control.

We have now inserted new data sets using a more appropriate control (SEZ6 Δ CYTO-ER-HA). Our control cells were transfected with this modified version of SEZ6 (SEZ6 Δ CYTO-ER-HA) that carries an ER-retention signal and is retained in the ER (new Fig. 6).

c. What is the criterion to identify "Golgi-derived vesicles", as stated in the legend for Fig. 6B? Again, co-exoression (or co-labeling) with a Golgi marker may give this information. Otherwise, white arrowheads in panel b are just pointing to vesicles.

We have now inserted images in which we co-stained SBP-GFP-GluK2 with the ER marker Calnexin at timepoint "0" of biotin addition (Suppl. Figure 5A, top row panel). We also used two Golgi markers to confirm the trafficking through the Golgi apparatus from cis-Golgi (GM130) to the trans-Golgi Network (TGN46) after 20 minutes after biotin addition, respectively (Suppl. Figure 5A, second and third row of panels). To further validate and test the identity of the GluK2 vesicles that we are counting, we co-transfected SBP-GluK2-GFP with SBP-LysozymeC-RFP. We observed a 50 % overlap of these vesicles, suggesting that these vesicle populations are transported in the same vesicles that start budding from the TGN after 20 minutes and further accumulate after 40 min after biotin addition. LysozymeC vesicles are sphingomyelin-rich and have been intensively characterized in previous studies (Deng et al., 2016, PNAS and Deng et al., 2018, Dev Cell). Also, it is essential to note that the vesicles that we count are only visible after 20 min after biotin addition.

d. In the Methods section, it is stated that "only cells showing transport of the reporter from the ER to Golgi after biotin incubation were considered, while cells showing ER signal after biotin addition were excluded". What does this mean? In the sample figures in panel b, there is plenty of red signal not present in vesicles. How can you determine that this is not ER? Overall, the quality of these images seems too low to make conclusions about compartment identities.

To exclude cells from an experiment is a valid procedure when applying the RUSH method and also reported in former publications (Deng et al., 2018). Investigating a pool of cells by immunofluorescence microscopy, there is always around 1% of cells in which the RUSH construct never leaves the ER. In 99% of the cells, however, the protein nicely moves from the ER to the Golgi after the addition of biotin. As stated above, we have now inserted a RUSH experiment in which we labeled the Golgi and ER, and TGN derived vesicles, respectively.

e. Do the authors extract (or can extract) any conclusion from the extent of colocalization between SBP-mCherry-GluK2 and SEZ6 (or SEZ6L2)?

Potentially, this type of images may give valuable information on the co-transport of SEZ6 with GluK2, but again, images of much better quality would be required.

We have generated more data investigating this point (Suppl. Figure 5C, D). New data clearly demonstrates that SEZ6 wild type vesicles partially colocalize with SBP-GluK2-GFP vesicles 40 minutes after biotin addition. The colocalization of both proteins in vesicles suggests that both are partially co-transported in the same vesicles emerging from the TGN. We have added a new figure panel (Suppl. Figure 5C) and plot with quantification (Suppl. Figure 5D) of colocalization.

4. The authors argue that HEK293 cells are used for the trafficking assay to avoid competition with endogenous protein. Still, given the specializations of neurotransmitter receptor trafficking in neurons, a trafficking assay of labeled GluK2/3 on wt or SEZ6 knockout neurons would be much more convincing. In fact, taking into account my comment #2, I consider that a more physiological assay in neurons, either for receptor trafficking or for the modulation of synaptic function, would significantly strengthen the mechanistic interpretation of the role of SEZ6 in the regulation of KAR function.

As suggested by the reviewer we added a more physiological assay in neurons for receptor trafficking of GluK2. We used the RUSH system, similar to our experiments in Fig. 6, where we had used HEK cells. Now, however, we analyzed primary neurons following a previous publication from the Henley lab (Evans et al. Cell Reports 2017) and quantified the ratio of surface GluK2 to total GluK2. First, we analyzed trafficking of the transfected GluK2 construct in a time-dependent manner and then chose the appropriate time point (20 min) for testing the influence of SEZ6 on GluK2 trafficking in neurons. Unlike in the HEK293 cells in Fig. 6, we did not overexpress SEZ6, but instead deleted the endogenous SEZ6 with lentiviral CRE transduction of floxed SEZ6 neurons. Our results (shown in new Fig. 7) are fully in line with the results obtained in Fig. 6 in HEK293 cells and demonstrate that the presence of endogenous SEZ6 (Ctrl. versus SEZ6 KO) enhances cell surface trafficking of GluK2. Description of the experiments is included in the results' section from page 9 (bottom) to page 11.

5. How were SEZ6 constructs expressed in neurons for Fig. 7 (panels B-D)? This is a very important experiment, as it evaluates the effect of SEZ6 mutants on the maturation of endogenous GluK2/3 in neurons. However, there is no information in the figure legend or the Methods section (it is just stated that neurons were "transduced"). In order to see changes in glycosylation of endogenous GluK2/3, the fraction of transduced cells had to be close to 100%. No viral vector is mentioned in the text, and plasmid transfection on neurons is typically far from 100% efficient.

We are sorry to not have described this method in detail. We made the following changes:

- A) The following sentence was added to the results on page 11 (bottom): "SEZ6 KO cortical neurons were infected the same day they were plated (DIV0) with the lentiviral constructs represented in Figure 8A and lysates were collected at DIV 7 for glycosylation analysis."
- B) In the methods section on page 18 (bottom), the paragraph "Molecular biology" was renamed to "Lentiviral plasmids". Description of the plasmids was left unchanged, but we included the reference to the original pFUGW vector.

We also listed the names of the VSV-G and gagpol plasmids for lentivirus production.

- C) On page 19 in the methods' section the lentiviral transduction was added: "In general, neurons were infected with lentiviruses expressing GFP, CRE or the different constructs described above, at day 2 *in vitro* (DIV) with the exception of the rescue experiment in figure 8 (infection was done at DIV 0 in order to maximize the rescue effect). After 5 days *in vitro* (DIV), neurons were washed with PBS, medium was replaced with fresh neurobasal media supplemented with B27 and the experiments were carried out at DIV 7."

Minor point.

6. I am not sure it is justified to state that SEZ6 cannot be considered an auxiliary protein for KARs because their co-immunoprecipitation from brain extracts was negative (Discussion). Many factors could affect the outcome of an immunoprecipitation, and it is always risky to extract conclusions from negative results.

We fully agree with the comment. Accordingly, we added the following sentences to the results' section on page 13, where the interaction experiment is described: "This suggests that the SEZ6 interaction with GluK2 and/or GluK3 is weaker or shorter-lived than the Neto2-GluK2/3 interaction and not stable enough to be detected in our extract conditions. Yet, it is possible that technical limitations prevented us from detecting an endogenous interaction of SEZ6 and GluK2. The successful demonstration of the interaction of two endogenous membrane proteins depends on the right choice of detergent. Thus, milder detergents than the NP40 used here may potentially allow the detection of endogenous SEZ6-GluK2 interactions."

Referee #3:

Summary: The authors report the results of proteomics from membranes of WT and SEZ6 KO mice. Given the relevance of SEZ6 in various disease the results of this study is important and of interest in the field. However, the major concerns are the lack of quantitative evidence for the conclusion drawn and the unconvincing nature of many of the immunoblots shown.

Major concerns:

Figure 3: Auxiliary factors such as SOL-1 and NETO alter biophysical properties of iGluRs. The authors should test if there are alterations in such properties in neurons and heterologous cells with co-expression of SEZ6.

This is an interesting and important point. None of the four labs with expertise in electrophysiology included in our study (Mulle/Bordeaux, Rammes/Munich, Köglspurger/Munich, Busche(Konnerth)/Munich) had a currently running fast application setup which, however, is required for analyzing the biophysical properties of GluK2. The Mulle lab will set it up again for HEK293 cells in the near future (when the labs start running again), so that we will certainly carry out this experiment in our future work. Regarding primary neurons, we feel that the suggested experiment cannot be realistically done because of the KAR activation being too slow - so that it will not be possible to activate all neuronal receptors at the same time - and because of the topographical spread of the receptors. Still, we are happy to give it try. We found now

a lab that would be willing to do this analysis, but this requires embryo transfer of our mice to the collaborator's lab. Because this will take between 6 months and one year, we will address these experiments in a follow-up study.

Consequently, we deleted the term "function" from the title of our manuscript, which may be misunderstood. Some readers may feel that function refers to biophysical properties, which we did not investigate in the manuscript. The shortened title is now: Seizure protein 6 controls glycosylation and trafficking of kainate receptor subunits GluK2 and GluK3".

Page 8: The statement, "Total levels of the GluK2/3 bands were not changed and this was also seen for a control protein, the GluA2 subunit of AMPA receptors (Fig. 4B and D)" is misleading because total levels were significantly decreased in neuronal lysates (Figure 4C) but not brain lysates (Figure 4D). Furthermore, quantification for GluA2 as control should be included in the graphs in Figures 4C and 4D.

As suggested, we clarified this point and additionally quantified GluA2 levels, both in the primary neurons and in the brain samples. The new sentence on page 8 is: "In contrast to the primary neurons, total levels of the GluK2/3 in the brain samples were not significantly decreased and this was also seen for a control protein, the GluA2 subunit of AMPA receptors (Fig. 4B and D, Suppl. Fig. 3C)". Additionally, we modified one sentence in the discussion on page 15, which is now: "loss of SEZ6 reduced neuronal surface levels of GluK2 and/or GluK3 in a post-transcriptional manner with only a minor effect on total protein levels of GluK2/3 in neurons and no effect in mouse brains."

Figure 4E: There appears to be a reduction in the level of SEZ6 in the SEZ6L2KO; thus, one might expect changes in the glycosylation level of GluK2/3, at least to a lesser extent, as shown in SEZ6 KO lysates (Fig. 4A). The blots shown in Figure 4E should be quantified to determine if there is a significant decrease in SEZ6, and if so, provide an explanation of the lack of glycosylation changes in GluK2/3.

As suggested we quantified SEZ6 in the synaptosome preparations in Fig. 4E and chose PSD-95 (which is also shown in the Western blots) for normalization. The quantifications are shown in new suppl. Fig. 3D and demonstrate that there is no significant reduction of SEZ6 in the SEZ6L2KO. This is also in line with a previous publication where SEZ6 was not altered in SEZ6L2KO brain extracts (Pigoni et al. Mol Neurodegen. 2016). This information is now included in the results section on page 8: "Although SEZ6 has two homologs, SEZ6L and SEZ6L2, which have a similar domain structure as SEZ6, there was no compensatory change in SEZ6 expression nor an effect on mature glycosylation of the GluK2/3 band in SEZ6L and SEZ6L2 single knock-out brain synaptosomes (Fig. 4E and Suppl. Fig. 3D)."

Figure 4F: The conclusion that, "demonstrating that the effect of SEZ6 on maturation of GluK2 and/or GluK3 is independent of the developmental stage and that no compensation occurs through other proteins during adulthood," is confusing as it would be expected that GluK2/3 maturation deficits would be dependent on SEZ6 level. In addition, the glycosylation pattern observed at E16.5 is very different from later ages and this pattern is dependent on SEZ6

level during developmental stage. This paragraph should be rewritten to accurately reflect the data shown and conclusions that can be drawn.

As suggested we rewrote this paragraph on page 8, which now reads as: "The relevance of SEZ6 for GluK2/3 maturation was not only seen at very young ages, when SEZ6 expression is high ((Kim et al., 2002, Miyazaki et al., 2006) and Fig. 4F), but also during adulthood (Fig. 4F). This demonstrates that SEZ6 has an effect on maturation of GluK2 and/or GluK3 independent of the developmental stage and that no compensation occurs through other proteins during adulthood. Interestingly, the lowest GluK2/3 band was more intense during the embryonic stage, and its intensity decreased sharply after birth (Fig. 4F). Considering that the reduction was equally occurring in WT and SEZ6KO mice, we conclude that this effect was dependent of the developmental stage, but independent of SEZ6."

Regarding SEZ6 levels: they are higher in embryonic brain, when SEZ6 appears to be more widely expressed than after birth, at which time point the expression of SEZ6 is restricted for example to the CA1 region within the hippocampus (Pigoni et al. Mol Neurodegen 2016). In contrast, GluK2 and 3 are more widely expressed. Thus, it is well possible that during embryonic development SEZ6 has a role on GluK2/3 trafficking in a larger number of neurons compared to the postnatal brain, where only a partial coexpression of GluK2 and SEZ6 is seen. This point is now included in the following paragraph on page 15 (discussion): "Importantly, in the absence of SEZ6, surface levels of GluK2 and GluK3 were reduced by about 50%, but were not completely abolished, indicating that GluK2 and GluK3 surface transport can – at least partly – happen in the absence of SEZ6. Other proteins may allow the remaining cell surface transport in the absence of SEZ6. Alternatively, it is possible that SEZ6 is required for GluK2 and GluK3 transport in only some, but not all neurons in the brain. In fact, while GluK2/GluK3 and SEZ6 are coexpressed in some areas of the brain, such as the CA1 region of the hippocampus and mid-cortical layers of the cortex (Foster et al., 1981, Kim et al., 2002, Straub et al., 2011), GluK2 and GluK3 are also expressed in other brain areas, such as the CA3 region of the hippocampus, where SEZ6 is barely expressed (Kim et al., 2002, Pigoni et al., 2016, Watanabe-Iida et al., 2016)".

Page 9: The authors state that, "So far, our experiments revealed that SEZ6 is required for normal cell surface localization and signaling as well as for correct glycosylation of GluK2 and/or GluK3 with the HNK-1 epitope." It is not clear to what 'signaling' refers, do the authors mean 'ion channel conductance' as shown in Figure 3? Please clarify this term. The term signaling is used throughout the discussion and should be clarified here as well.

With the term signaling we referred to kainate-evoked KAR currents. As suggested, we now exchanged the expression 'signaling' and use instead kainate-evoked KAR currents or, alternatively deleted the expression signaling and only refer to surface glycosylation and glycosylation. As an example, the sentence referred to by the reviewer therefore now reads as: "So far, our experiments revealed that SEZ6 is required for normal cell surface localization and correct glycosylation of GluK2 and/or GluK3 with the HNK-1 epitope." (page 9).

Figure 6: The images shown are not particularly convincing. What is the green immunofluorescence? The authors state that, "SBP-mCherry-GluK2 was retained in the ER (Fig. 6B)" without use of any ER marker to validate this statement at the 0 min time point. Similarly, no Golgi marker is used to confirm forward trafficking to this organelle at 20 min. Furthermore, the authors state that

cytoplasmic vesicles appear at the 40 min time point but there are also vesicular structures pointed out and quantified in the 20 min time point. Also, no explanation as to how 'vesicles' are quantified is provided in the methods.

Thank you for pointing this out. Because Reviewer 2 raised the same points, we have provided a completely new Figure 6 with new experiments and more details, including a new Fig. 7 and a new suppl. Fig. 5. Also, we split the channels, and labeled the images correspondingly (**Figure 6B and 6C**). We provide additional information in the figure legends (page 39) and inserted details on antibodies and fluorophores in the updated version of the Materials and Methods section on pages 27-28. Within the results' section, the new experiments are described from page 9, bottom, to page 11.

We carried out experiments in HEK293 cells (new Fig. 6 and new suppl. Fig. 5) and also in primary neurons (new Fig. 7), where the endogenous SEZ6 was deleted with the help of lentiviral CRE transduction in floxed SEZ6 neurons.

In the HEK cells (suppl. Fig. 5A), we did costaining of GluK2 with markers for ER, Golgi, TGN and vesicle. These images clearly show that in the absence of biotin, SBP-GFP-GluK2 colocalizes with the ER marker Calnexin (Figure 5A, first panel). 20 minutes after addition of biotin to the cells trafficking of SBP-GFP-GluK2 was observed to the Golgi apparatus (staining for GM130) and partly to the TGN (TGN46 staining) where GluK2 is packed into specific vesicles (Figure 5A, third panel). Importantly, these vesicles start to emerge 20 min after biotin addition (before they are not visible). To prove the identity of the counted spots as vesicles, we co-transfected SBP-GluK2-GFP and SBP-LysozymeC-RFP and imaged vesicles emerging from the TGN 40 min after biotin addition. We observed a substantial overlap between SBP-GluK2-GFP and SBP-LysozymeC-RFP suggesting that both proteins are cotransported. LysozymeC vesicles are sphingomyelin-rich vesicles that have been well characterized in prior studies (Deng et al., 2016, PNAS and Deng et al., 2018, Dev Cell). For quantification of these occurring cytosolic vesicles, we empirically measured the sizes of objects between 4 and 20 pixels using the Analyze Particles function in ImageJ, which detects vesicular structures but omits larger structures such as the Golgi. While small-fragmented and isolated Golgi structures could be detected in error, such structures are rare. Colocalization and quantification are shown in new suppl. Fig. 5C.

Figure 7: It is difficult to judge 'rescue' of GluK2/3 maturation based on the stated 'reappearance' of the uppermost band. The authors should perform biotinylation studies as in Figure 2A which allow for quantification of the extent of rescue. The results of the glycosylation state could then be interpreted in that context. It is surprising that SEZ6 Δ Cyto was expressed at significantly lower intensity on the blot and yet could 'rescue' the upper mature band for GluK2/3 and should be explained. Quantification of the 'rescue' might help to understand the lack of relationship between expression level of the constructs and surface expression of GluK2/3. Furthermore, the blot shown in Figure 7D is not convincing to demonstrate that SEZ6/delta-cyto can 'rescue' the mature glycosylation compared with SEZ6/delta-cytoER, even with the apparent longer running of the gel as evidenced by the increase area between the upper and

lower bands of SEZ6 in Figure 7D compared with Figure 7B. Furthermore, the IP results shown in Figure 7E require explanation as to the significance of the HA.11 signal just above the 25 kDa molecular weight marker. It also appears that co-expression of the SEZ6 constructs, except for the CTF, negatively impacts the level of GluK2/3 in HEK293 cells.

This paragraph raises several points so that there are six parts to our answer.

- A. As suggested by the reviewer, we performed a biotinylation experiment in primary SEZ6KO neurons lentivirally transduced with either GFP (as a negative control maintaining the SEZ6KO state) or with full-length SEZ6 (rescuing SEZ6 expression). As shown in new Suppl. Fig. 6 wild-type SEZ6 increased surface expression of GluK2/3 by nearly two-fold, which is in excellent agreement with the opposite experiment in Fig. 1 and 2, where the knock-out of SEZ6 had reduced GluK2/3 levels by about 50%.
- B. The reviewer would like to see whether the different SEZ6 mutants have a quantitatively different rescue ability. In the surface biotinylation experiment in Suppl. Fig. 6, we noted that the rescue experiment was accompanied by a much larger variation (Suppl. Fig. 6) compared to the knock-out experiments in Fig. 1 and 2, presumably because the lentiviral efficiency for reintroducing SEZ6 into the SEZ6KO neurons was less efficient and more variable (among the individual cells) than transducing floxed SEZ6KO neurons with Cre (Fig. 1 and 2), where even minor amounts of the enzymatically active Cre are sufficient to achieve SEZ6 knock-out in the floxed neurons. Given the relatively small effect size (two-fold increase) combined with the larger variability, we realized that the surface biotinylation is not a suitable way to address the reviewer's points, namely the quantification of the degree of rescue exerted by the different SEZ6 mutants. For this reason, we included surface biotinylations only for full-length SEZ6 (Suppl. Fig. 6), but did not carry them out for the SEZ6 mutants. We realize that for our future follow-up work, we need to develop a different sort of assay, that would be more sensitive for potentially subtle changes among the mutants.
- C. Yet, in order to get some quantification, we plotted the intensity distribution of the three GluK2/3 bands as a histogram (Suppl. Fig. 3A). Compared to the immunoblots this data representation allows to see more easily which constructs rescued the glycosylation phenotype of the SEZ6KO neurons (for a detailed description see page 24 (bottom) in the methods section and page 8 in the results section). Because the difference between WT and SEZ6KO is the presence or absence of the uppermost GluK2/3 band, we would ideally only quantify this uppermost band. Yet, this is not possible, because this band runs too close to the next GluK2/3 with a slightly lower molecular weight. Thus, we quantified the differences between WT and SEZ6KO in a different way. We divided the histogram into quartiles and focused on the fourth quartile, corresponding to the highest molecular weight in the immunoblot, where the differences are seen between WT and SEZ6KO. Mean intensities of the fourth quartile were calculated and were significantly smaller for SEZ6KO compared to WT (Suppl. Fig. 3A), in agreement with the immunoblot. This quantification assay was then used for the rescue experiment in Fig. 8B and D. SEZ6 FL, Dcyto and ecto rescued the glycosylation defect – as seen by the increase in mean intensity in the fourth quartile. In contrast, SEZ6CTF did not rescue. Thus, the quantification assay is well suited for detecting whether rescue occurred or not. While all three constructs (SEZ6 FL, Dcyto and ecto) rescued, there was no significant difference among them. Thus, either all three

constructs rescue to the same extent, or the assay is not sensitive enough to detect differences among them. Thus, at this point we can conclude whether a mutant SEZ6 form has the ability to rescue the SEZ6KO phenotype or not, but we are not able to tell whether one mutant rescues more or less than the other ones (described in the results on page 8 and 12). Thus, either the mutants rescue as well as the wild-type, full-length SEZ6 or, otherwise, for future mechanistic studies we first need to develop an assay that is more suitable for detecting small changes between the mutants.

- D. The reviewer wondered, why some mutants (e.g. SEZ6 Δ cyto) – although being expressed at lower levels than SEZ6 FL – rescued the glycosylation phenotype to the same extent as SEZ6 FL. We observed that the SEZ6 FL lentivirus led to a 5-10 higher expression of the exogenous SEZ6 compared to the endogenous SEZ6 in wild-type neurons (s. figure for reviewer below).

Figure for reviewer: Comparison of SEZ6 expression under endogenous and transduced conditions. Endogenous SEZ6 runs as two bands (immature - marked with arrow - and mature, fully glycosylated, lanes 1, 2), but is not seen upon CRE-mediated SEZ6 knock-out (lanes 3, 4) or in SEZ6KO neurons transduced with control (GFP) lentivirus (lanes 5, 6). Lentiviral trans-duction of SEZ6KO neurons (lanes 7, 8) restores SEZ6 expression. Note that the transduced full-length SEZ6 (SEZ6FL) levels are much higher than the endogenous ones. Transduced SEZ6 has a higher apparent molecular weight than under endogenous conditions because of extra epitope tags and linkers cloned into the lentiviral construct.

Thus, we expect that a lower titer of this virus would have been sufficient to rescue the glycosylation phenotype. Therefore, we are not surprised to see that the lower SEZ6 Δ cyto expression in Fig. 8B was sufficient to rescue the glycosylation phenotype of GluK2/3 as well.

- E. The intensive band above 25 kDa in the HA.11 blot in panel 8E (second panel from top) is the light chain of the GluK2/3 antibody used for immunoprecipitation. This is why this band shows up in all lanes, because the immunoprecipitation was done for all samples, including the negative controls. This is now indicated in the figure legend and the band is marked with an asterisk.
- F. The reviewer noted that from panel 8E (lowest bands for GluK2/3) it appears that the three SEZ6 constructs reduce expression pattern of GluK2/3. We feel that this is a technical artefact of the transient cotransfection of the SEZ6 constructs together with the GluK2/3 construct. In such situations we often observe that strong expression of one construct interferes with efficient expression of another construct, probably due to competition of transcription and/or translation factors. The reason why SEZ6CTF would not show the same effect is that SEZ6CTF is a very small protein, whereas the other constructs encode large proteins, where competition of transcription/translation factors is likely to be more relevant. In any case, the results in

panel 8B demonstrate that the very same constructs have no effect on the expression level of endogenous GluK2/3, ruling out the possibility that the expression changes in 8E are physiologically meaningful in neurons.

Minor concerns:

Introduction page 5: Neto binds Kainate receptors not AChR or AMPAR as stated.

The text was changed as suggested. The new text on pages 3 and 4 now reads as: "Both CUB and CCP domains are frequently found in proteins of the complement system (Escudero-Esparza, Kalchishkova et al., 2013, Forneris, Wu et al., 2016), but also in auxiliary subunits of neurotransmitter receptors, for example LEV-10 for acetylcholine receptors (AChR) and Neto2 for kainate receptors (Gally, Eimer et al., 2004, Zhang, St-Gelais et al., 2009)."

Page 7: The conclusion that, "Taken together, these experiments suggest that SEZ6 controls cell surface levels of GluK2/3 at the post-translational level," should be updated to include reference to the modest effect on total GluK2/3 levels (15% reduction). For example, "Taken together, these experiments suggest that SEZ6 primarily controls cell surface levels of GluK2/3 at the post-translational level; however, a modest effect on total levels, indicating reduced biosynthesis, was also observed."

This suggestion was incorporated into the results' section. The new sentence on top of page 7 is:

"Taken together, these experiments suggest that SEZ6 primarily controls cell surface levels of GluK2/3 at the post-translational level, while only moderately affecting total protein GluK2/3 levels".

Page 11: The statement that, "suggesting that under physiological conditions it is preferentially the full-length form of SEZ6 that controls SEZ6 glycosylation and cell surface levels". - this should be 'controls GluK2/3 glycosylation and cell surface levels.'

This typo was corrected – now on page 13.

Figure 4A: The arrows in the panel at right are distracting and should be replaced with asterisks as in the EndoH portion of the panel. The indication 'partially EndoH resistant' is confusing, to what is this arrow referring? There is a purple asterisk that seems to refer to a blue band, which is probably supposed to also be purple. Furthermore, the asterisks in the SEZ6 blot (upper left) are not explained.

We are sorry that the previous version of the figure was not clear enough. As suggested we made the model in Fig. 4A clearer. We removed the arrows as well as the description of 'partially EndoH-resistant', as this is clearly explained now in the results description. We also chose a new color coding for the individual bands. The asterisks on the blot in panel Fig. 4A were removed. They had referred to the immature form of SEZ6, which shows the expected decrease in molecular weight after EndoH-treatment, whereas the mature, fully glycosylated form of SEZ6 (upper, more intensive band) does not show a molecular weight shift, because it is EndoH-resistant. Because the intensity of this band is low, we removed this information for clarity reasons.

Figure 4B: The glycosylation patterns in Figures 4A and 4B appear different; thus, it would be useful to include a zoomed-in panel as shown for Figure 4A.

We fully agree with the observation of the reviewer. In the primary neurons (Fig. 4A) there are three bands of GluK2/3 visible after EndoH treatment, whereas only 2 bands are visible in Fig. 4B for whole brain extracts. This difference is also seen in Fig. 4F, where the band pattern for GluK2/3 is shown at different developmental ages. Because the neurons were prepared from E16.5 old mouse embryos, their glycosylation pattern corresponds to the E16.5 time point in Fig. 4F, whereas the glycosylation in Fig. 4B corresponds to the postnatal age in Fig. 4F.

As suggested we included a zoomed-in panel as for Fig. 4A, and show it now as new suppl. Fig. 3B.

Figures 5-7: It is not stated how many replicates of the glycosylation or immunoprecipitation experiments were performed in the legend or methods.

As suggested this information is now included in the legends. For example, in the legend to Fig. 5 it is now written: "HNK-1 band on GluK2/3 was detected only in WT brains but not in SEZ6KO brains (n=6). As a control, HNK-1 modification of NCAM-1 was unaltered, as revealed after IP of NCAM-1 from WT and SEZ6KO brains (n = 3), followed by detection with anti-HNK-1 antibody. Different isoforms of NCAM-1 are annotated with arrowheads (NCAM 180, NCAM 140 and NCAM 120). *Mid panel:* To prove the specificity of the HNK-1 antibody, GluK2/3 was immunoprecipitated and digested with Peptide-N-Glycosidase F (PNGase F), which removes N-linked oligosaccharides. Upon PNGase F digestion, HNK-1 was not detectable in WT brains and the molecular weight of GluK2/3 was reduced (n=2)."

And in the legend to Fig. 8C (previously 7C) it is now written: "Representative blot of 3 independent experiments."

Dear Stefan,

Thank you for submitting your revised manuscript to The EMBO Journal. Your study has now been seen by the referees # 2 and 3 and their comments are provided below. Both referees appreciate the introduced changes and support publication here. I am therefore very pleased to let you know that we can accept the manuscript for publication here. Before we can formally accept the manuscript for publication here there are just a few things to sort out:

- Please address Ref 2's point regarding Fig 8A
- COI statement is missing
- Author contributions are missing
- Please add the legend for supplemental table 1 as a separate tab in the excel file and remove the legend from the main manuscript file. Make sure you use proper call out to this table in the MS text - see also guide to authors
- The supplemental file need to be re-labeled as an appendix and needs a ToC. The legends to the supplemental figures need to be included in the appendix and removed from main MS file. Please also make sure that you use the correct nomenclature - see guide to authors. Just contact us if you have questions regarding this. Can you also make sure that the resolution of the figures in this file is OK. Some of the figures look a bit pixelated when you zoom see for example supplemental figures 5 and 6. For Supplemental Fig 6a it would also be good to see the source data as it is unclear if there is a splice site in the Flag lane.
- We also require a Data Availability section. As your proteomics data is provided in the MS files - you can simply say this study includes no data deposited in external repositories or something like this.
- In general, we encourage the publication of source data, particularly for electrophoretic gels and blots, with the aim of making primary data more accessible and transparent to the reader. It would be great if you could provide me with a PDF file per figure that contains the original, uncropped and unprocessed scans of all or key gels used in the figure? The PDF files should be labeled with the appropriate figure/panel number, and should have molecular weight markers; further annotation could be useful but is not essential. The PDF files will be published online with the article as supplementary "Source Data" files.
- Our publisher has also done their pre-publication check on your manuscript. When you log into the manuscript submission system you will see the file "Data edited manuscript file". Please take a look at the word file and the comments regarding the figure legends and respond to the issues. Please also use this version when you resubmit the revised version with the marked changes. Just makes it easier for us to see the changes.
- We include a synopsis of the paper (see <http://emboj.embopress.org/>). Please provide me with a general summary statement and 3-5 bullet points that capture the key findings of the paper.
- We also need a summary figure for the synopsis. The size should be 550 wide by 400 high (pixels).

You can also use something from the figures if that is easier.

I think that should be it. You can use the link below to upload the revised version.

Let me know if you have any further questions.

with best wishes

Karin

Karin Dumstrei, PhD
Senior Editor
The EMBO Journal

Further information is available in our Guide For Authors:

The revision must be submitted online within 90 days; please click on the link below to submit the revision online before 9th Aug 2020.

Link Not Available

Referee #2:

The authors have been very responsive and have added new experimental evidence to address the comments of the reviewers. Particularly, the new assays to monitor intracellular trafficking of GluK2, surface biotinylation and the RUSH assay to quantify surface trafficking in neurons provide a much more convincing evidence for the effect of SEZ6 on GluK2. Also, further methodological information has been included regarding the expression of recombinant proteins in neurons and the identification of intracellular components by immunostaining.

I am also satisfied with the authors toning down their references to the physiological functions of SEZ6 in neurons. Their work is focused on intracellular trafficking of kainate receptors, and this topic is important enough without the need to oversell it.

Overall, my concerns on the previous version of this manuscript have been properly addressed, and I now consider that this work is suitable for publication.

Minor point: in Fig. 8A, the delta symbol appears to be missing from the names of the truncated SEZ6 mutants.

Referee #3:

In this revised manuscript the authors report the results of proteomics from membranes of WT and SEZ6 KO mice and demonstrate that SEZ6 controls glycosylation of GluK2/3 and surface trafficking. Given the relevance of SEZ6 in various disease states the results of this study are important and of interest in the field. The authors have addressed the concerns raised in the previous review.

Here is our point-by-point response to the open questions:

- Please address Ref 2's point regarding Fig 8A
The delta symbols were added.

- COI statement is missing
This was added after the acknowledgement.

- Author contributions are missing
This was added after the acknowledgement.

- Please add the legend for supplemental table 1 as a separate tab in the excel file and remove the legend from the main manuscript file. Make sure you use proper call out to this table in the MS text - see also guide to authors

Because the legend was anyway short, we added it directly on the same tab – on the very top -, where all the data are and did not put it into a separate tab. We feel that in this way it is easier for the reader to quickly get the information.

- The supplemental file need to be re-labeled as an appendix and needs a ToC. The legends to the supplemental figures need to be included in the appendix and removed from main MS file. Please also make sure that you use the correct nomenclature - see guide to authors. Just contact us if you have questions regarding this. Can you also make sure that the resolution of the figures in this file is OK. Some of the figures look a bit pixelated when you zoom see for example supplemental figures 5 and 6. For Supplemental Fig 6a it would also be good to see the source data as it is unclear if there is a splice site in the Flag lane.

We made the required changes. To conform to EMBO style, we also selected five figures (among the figures previously labeled as supplementary figures) and now include them as EV figures. The remaining ones are now labeled as Appendix figures and the appendix is now organized as required.

We hope that the resolution of our new PDF documents is now sufficient. For all figures we now provide high resolution PDFs. If needed, we are happy to provide the files also as EPS-files.

For the figure previously called Suppl. Fig. 6a (now Appendix Figure S3) we added pictures of the whole membranes as source data (see also your suggestion below, which we followed, so that we now provide pictures of the key blots from the different figures).

- We also require a Data Availability section. As your proteomics data is provided in the MS files - you can simply say this study includes no data deposited in external repositories or something like this.

This was added after the acknowledgement.

- In general, we encourage the publication of source data, particularly for electrophoretic gels and blots, with the aim of making primary data more accessible and transparent to the reader. It would be great if you could provide me with a PDF file per figure that contains the original, uncropped and unprocessed scans of all or key gels used in the figure? The PDF files should be labeled with the appropriate figure/panel number, and should have molecular weight markers; further annotation could be useful but is not essential. The PDF files will be published online with the article as supplementary "Source Data" files.

As suggested we added the key blots. Some of them appear relatively light in color. But you can click on them, which highlights them in blue and allows you to even see the very faint bands on these blots.

- Our publisher has also done their pre-publication check on your manuscript. When you log into the manuscript submission system you will see the file "Data edited manuscript file". Please take a look at the word file and the comments regarding the figure legends and respond to the issues.

Please also use this version when you resubmit the revised version with the marked changes. Just makes it easier for us to see the changes.
We addressed all of the points.

- We include a synopsis of the paper (see <http://emboj.embopress.org/>). Please provide me with a general summary statement and 3-5 bullet points that capture the key findings of the paper.
This is now added.

- We also need a summary figure for the synopsis. The size should be 550 wide by 400 high (pixels). You can also use something from the figures if that is easier.
This is now added.

Dear Stefan,

Thank you for submitting your manuscript to The EMBO Journal. I have now had a chance to take a careful look at everything and appreciate the introduced changes. I am therefore very pleased to accept the manuscript for publication here.

Congratulations on a nice study!

with best wishes

Karin

Karin Dumstrei, PhD
Senior Editor
The EMBO Journal

Please note that it is EMBO Journal policy for the transcript of the editorial process (containing referee reports and your response letter) to be published as an online supplement to each paper. If you do NOT want this, you will need to inform the Editorial Office via email immediately. More information is available here: http://emboj.embopress.org/about#Transparent_Process

Your manuscript will be processed for publication in the journal by EMBO Press. Manuscripts in the PDF and electronic editions of The EMBO Journal will be copy edited, and you will be provided with page proofs prior to publication. Please note that supplementary information is not included in the proofs.

Should you be planning a Press Release on your article, please get in contact with embojournal@wiley.com as early as possible, in order to coordinate publication and release dates.

If you have any questions, please do not hesitate to call or email the Editorial Office. Thank you for your contribution to The EMBO Journal.

** Click here to be directed to your login page: <http://emboj.msubmit.net>

Corresponding Author Name: Stefan Lichtenthaler

Journal Submitted to: EMBO J

Manuscript Number: EMBOJ-2019-103457